# RACA-CLIP: Relation-Aware Compositional Alignment for CLIP

## Abstract

Vision-Language Models (VLMs) such as CLIP excel at broad multimodal tasks, yet struggle with **compositional reasoning**. Despite capturing coarse correlations, they often act like "bags-of-words" missing fine-grained structures such as **object–attribute bindings** and **inter-object relations**. We attribute this to: (i) limited compositional diversity in large-scale image–text data, and (ii) contrastive objectives that emphasize global alignment over grounded structure. To address this, we propose a **hierarchical fine-grained alignment framework** that explicitly bridges visual and textual components at the object, attribute, and relation levels. Unlike prior work relying on parsers, we leverage existing **scene graph annotated dataset** for structured supervision, without collecting any extra manual labels or annotating new images. We introduce a **hierarchical fine-grained loss** to complement standard contrastive learning by grounding entities and relations across modalities. Experiments on compositional benchmarks **SugarCrepe**, **What'sUp**, and **Cola** show large gains in capturing nuanced structure, while preserving performance on standard vision-language tasks. **RACA CLIP** method improves compositional reasoning accuracy by **+24.86% on SugarCrepe**, **+5.7% on What'sUp**, and **+4.76 on Cola**, offering a simple yet effective path toward stronger, human-like compositional understanding.

## 1 Introduction

Humans effortlessly integrate information across modalities, binding objects, attributes, and relations into coherent scene representations that generalize compositionally. For example, when shown an image of "a brown dog chasing a white frisbee," people instantly understand *who is chasing whom*, can rephrase it linguistically in multiple ways, and can generalize to novel scenarios such as "a white dog chasing a brown frisbee." This ability to flexibly compose known concepts into new configurations is a cornerstone of perception, reasoning, and language. It underlies skills such as recognizing novel concept combinations, performing fine-grained visual discrimination, or engaging in spatial and relational reasoning capabilities that humans display naturally and without explicit supervision. By contrast, modern machine learning systems, even when trained on billions of image–text pairs, struggle to replicate this compositional competence. Although they excel at coarse-grained alignment and exhibit impressive zero-shot performance on classification and retrieval benchmarks (Radford et al., 2021; Jia et al., 2021; Mu et al., 2021; Li et al., 2022b; Xie et al., 2023; Zhai et al., 2023; Chen et al., 2024), they often fail when asked to disambiguate attribute bindings, relational roles, or structural variations in otherwise similar scenes.

At the heart of most VLMs lies a *contrastive function*, models are trained to maximize similarity between paired image–text embeddings while minimizing similarity with mismatched pairs. This global contrastive objective, popularized by CLIP (Radford et al., 2021), underpins the remarkable zero-shot recognition and retrieval abilities of VLMs. However, because it operates on *single global embedding*, models often exploit shortcuts such as object co-occurrence while ignoring fine-grained bindings between attributes, objects, and relations(Abbasi et al., 2025). As a result, contrastive alignment struggles with *hard negatives* that differ via subtle attribute or relational edits. One mitigation is to fine-tune with such *hard negatives* (Yuksekgonul et al., 2023b; Sahin et al., 2023a; Doveh et al., 2023; Wang et al., 2024), but this still penalizes at the global level and lacks explicit supervision on *where* the difference lies leading to instability and poor localization (Fig. 1).

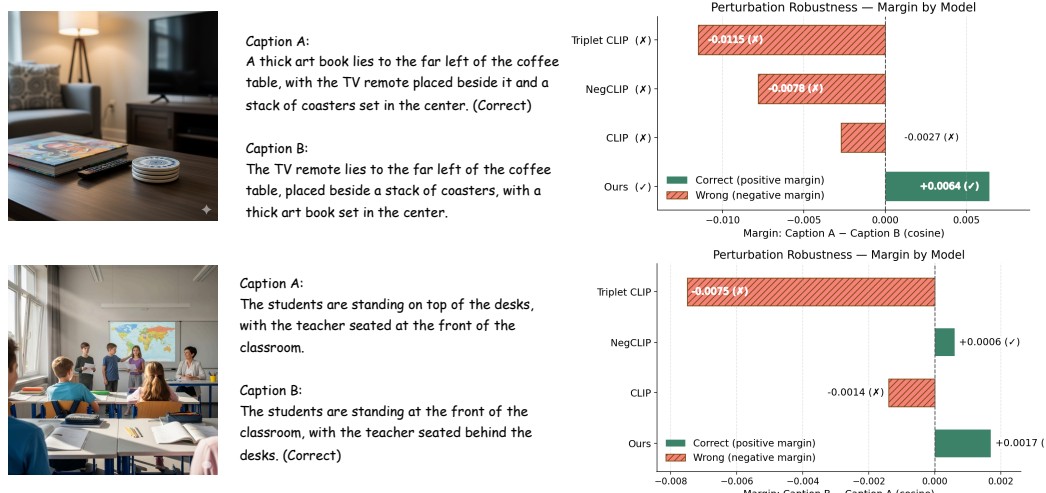

Figure 1: **Compositional robustness of CLIP variants.** Two multi-object scenes are paired with a ground-truth caption and a perturbed variant. We report cosine *margins* for each model, green indicates a correct preference ($m > 0$), red indicates a failure ($m < 0$). **RACA-CLIP (Ours)** consistently shows the highest margins, especially for subtle attribute or relation changes, reflecting improved fine-grained alignment. More examples in Appendix E.11

Scaling models and datasets hasn't resolved this limitation, performance plateaus on compositional benchmarks despite more data and parameters (Kamath et al., 2023; Ray et al., 2023a). This points to a deeper issue, *the contrastive function lacks structural inductive bias*. In contrast, *scene graphs* provide structured representations of entities, attributes, and relations (Johnson et al., 2015; Kim et al., 2024; Xu et al., 2024; Dutta et al., 2025), offering a natural scaffold for compositional alignment and are increasingly practical via modern parsers and detectors.

**In this work, we propose a structured contrastive framework that integrates scene-graph representations into CLIP models.** Our method: (1) aligns image regions with corresponding text descriptions via *IoU-based multi-positive matching*; (2) extends contrastive learning with a *dynamic, IoU-weighted contrastive function* that balances local and global signals; (3) introduces a *relation-aware contrastive loss* over subject relation object triplets, training on $\langle s, r, o \rangle$ structures to make models explicitly sensitive to relational semantics.

We evaluate on compositional benchmarks (SugarCrepe (Hsieh et al., 2023), What'sUp (Kamath et al., 2023), UT-Zappos (Isola et al., 2015) and Cola (Ray et al., 2023b)) and standard multimodal tasks (zero-shot classification and retrieval). RACA-CLIP consistently improves compositional reasoning while preserving and often enhancing global alignment performance. The key contributions of the paper can be listed as:

- **Region-level alignment.** A *parameter-efficient*, IoU-weighted contrastive loss aligns image sub-regions with matching text spans, improving object–attribute binding and mitigating center bias.
- **Relation sensitivity.** A *relation-aware* triplet loss over $\langle s, r, o \rangle$ enhances grounding by encoding roles and word order.
- **Comprehensive gains & analysis.** State-of-the-art results on **SugarCrepe**, **What'sUp**, **SugarCrepe++**, **Winoground**, **MMVP-VLM** and **Cola**, and detailed representation analysis on **UT-Zappos** showing improved disentanglement and stability.

We hope this work encourages a shift from purely global contrastive learning toward *structure-aware alignment*, bringing VLMs closer to human-like compositionality.

## 2 RELATED WORK

Contrastive dual-encoder pretraining at web scale exemplified by CLIP and ALIGN has become the dominant recipe for vision-language alignment, learning transferable features by pulling paired image–text embeddings together while pushing apart negatives (Radford et al., 2021; Jia et al., 2021;

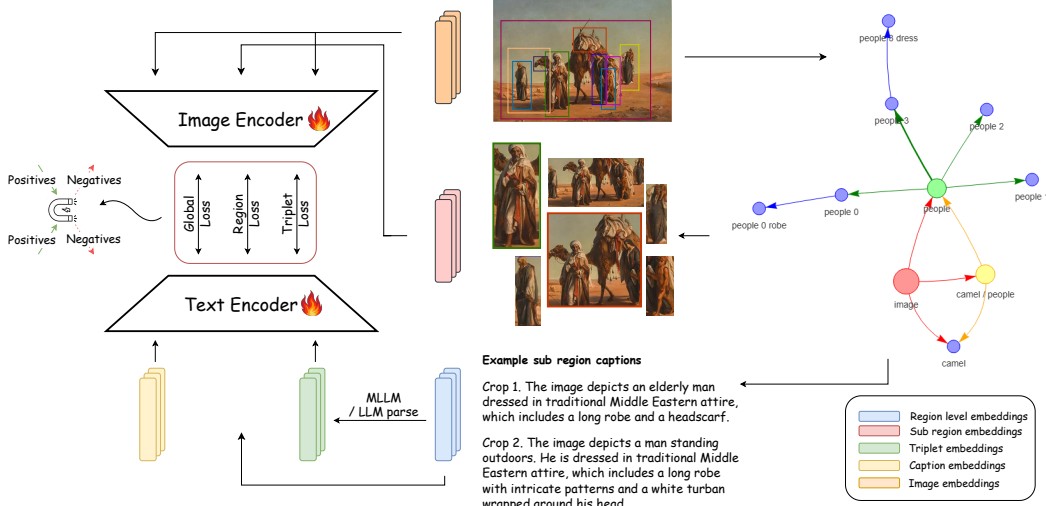

Figure 2: **Structured contrastive at a glance.** An image and caption are decomposed into *regions* and a scene graph. Three complementary losses act jointly: *global* image–caption alignment ($\lambda_g$), *region-level* IoU-weighted multi-positive alignment ($\lambda_r$), and *relation-aware* triplet alignment ($\lambda_t$). Structured outputs: sub-regions with IoU $\geq \tau$ treated as multi-positives (green), and a highlighted $\langle s, r, o \rangle$ path enforcing role/word-order sensitivity.

Schuhmann et al., 2022). This paradigm underpins strong zero-shot recognition and serves as a backbone for downstream multimodal systems (Mokady et al., 2021; Li et al., 2022a; Alayrac et al., 2022), yet its global-only objective often encourages shortcuts that miss fine-grained bindings and word order (Thrush et al., 2022; Yuksekgonul et al., 2023b). A range of extensions data filtering/augmentation, synthetic captions, and auxiliary losses help but do not eliminate these issues (Wang et al., 2024; Sahin et al., 2023b; Sinha et al., 2021; Li et al., 2023; Goel et al., 2022). Benchmarks that probe compositional generalization and spatial reasoning (What'sUp, Cola, SugarCrepe) consistently reveal near-chance behavior and "bag-of-objects/words" biases (Thrush et al., 2022; Yuksekgonul et al., 2023b; Ma et al., 2023; Kamath et al., 2023; Ray et al., 2023b; Hsieh et al., 2023). Hard negatives based minimal edits to attributes, relations, or word order help discrimination (Yuksekgonul et al., 2023b; Wang et al., 2024; Rösch et al., 2024) but can destabilize global alignment when applied solely at the embedding level (Kamath et al., 2024); related efforts target negation and unimodal composition (Alhamoud et al., 2025; Koishigarina et al., 2025). Motivated by these limitations, we retain the contrastive paradigm but inject explicit structure via scene-graph supervision: region-phrase alignment with IoU-weighted multi-positive contrast and relation-aware losses over $(s, r, o)$ triplets, providing the inductive bias needed for attribute binding and relational understanding while preserving global alignment (Zeng et al., 2022; Zhao et al., 2023; Kamath et al., 2023). A comprehensive survey and additional comparisons are present in Appendix B.

## 3 METHODOLOGY

In this section, we introduce our structured contrastive framework to improve compositional reasoning in CLIP-style models. Figure 2 presents an overview of the proposed framework. We reformulate image–text alignment through scene-graph decomposition (Hsieh et al., 2025), which balances the sparse supervision of captions with the rich structure of visual scenes. This motivates two new objectives: a region-level contrastive loss that aligns image sub-regions with their corresponding textual descriptions, and a relation-aware loss that captures object relation object triplets, addressing well-documented limitations of VLMs (Thrush et al., 2022; Hsieh et al., 2023; Kamath et al., 2023).

### 3.1 PRELIMINARIES: CLIP-STYLE CONTRASTIVE LEARNING

CLIP-style vision-language models adopt a dual-encoder architecture. A vision encoder $f_\theta(\cdot)$ maps an input image $I_i$ into a dense embedding $\mathbf{v}_i \in \mathbb{R}^d$ ( obtained from the [CLS] token of a Transformer

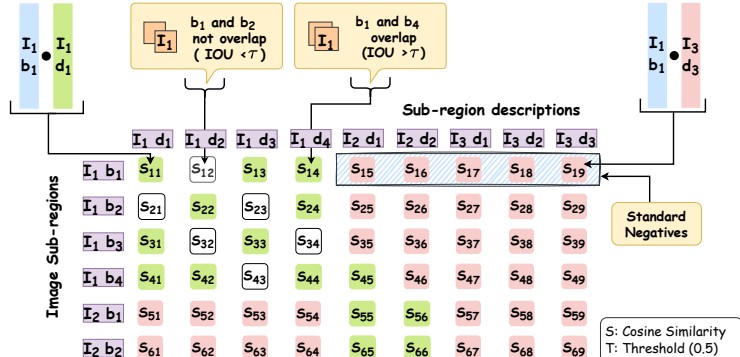

Figure 3: **IoU-weighted region–phrase contrastive loss.** A grid of image sub-regions is compared to phrase descriptions via cosine similarity $S$. Sub-regions with $\text{IoU} > \tau$ are treated as multi-positives and *weighted by their IoU*; sub-regions with $\text{IoU} \leq \tau$ have their intra sub-region similarities *masked* (excluded from the positive set), while remaining pairs act as negatives. This IoU-aware masking and weighting sharpen local grounding and yield more consistent attribute–object bindings.

encoder), while a text encoder $g_\phi(\cdot)$ maps the paired caption $T_i$ into an embedding $\mathbf{t}_i \in \mathbb{R}^d$ (using the final [EOS] token representation). Both embeddings are normalized onto the unit sphere:

$$\mathbf{v}_i = \text{norm}(f_\theta(I_i)), \qquad \mathbf{t}_i = \text{norm}(g_\phi(T_i)).$$

The similarity between the embeddings of an image $I_i$ and its caption $T_i$ is measured as

$$\text{sim}(\mathbf{v}_i, \mathbf{t}_i) = \mathbf{v}_i^\top \mathbf{t}_i.$$

Fineis performed with a symmetric contrastive objective (InfoNCE) over a batch of $N$ image-text pairs. For each image, the model is encouraged to retrieve its paired caption over all captions in the batch (*image → text* direction), and symmetrically, for each caption, the model must retrieve its paired image (*text → image* direction). The loss is:

$$\mathcal{L}_{\text{global}} = -\frac{1}{N} \sum_{i=1}^{N} \left[ \underbrace{\log \frac{\exp(\text{sim}(\mathbf{v}_i, \mathbf{t}_i)/\tau)}{\sum_{j=1}^{N} \exp(\text{sim}(\mathbf{v}_i, \mathbf{t}_j)/\tau)}}_{\text{image} \to \text{text}} + \underbrace{\log \frac{\exp(\text{sim}(\mathbf{t}_i, \mathbf{v}_i)/\tau)}{\sum_{j=1}^{N} \exp(\text{sim}(\mathbf{t}_i, \mathbf{v}_j)/\tau)}}_{\text{text} \to \text{image}} \right], \quad (1)$$

where $\tau$ is a learnable temperature parameter. This framework has proven highly effective for large-scale multimodal pretraining and underpins the success of CLIP (Radford et al., 2021).

## 3.2 INTER-MODALITY STRUCTURE VIA SCENE GRAPHS

Web-scale VLM pretraining data is *text-sparse* but *image-rich*: captions often omit attributes/relations that are visually present (Kamath et al., 2023). This mismatch encourages shortcut solutions. Scene graphs (Hsieh et al., 2025) reduce this imbalance by decomposing both modalities into fine-grained, aligned units.

**Notation.** For a caption $T$, let its scene graph be $\mathcal{G}^{\text{text}} = (\mathcal{O}, \mathcal{A}, \mathcal{R})$ with objects $\mathcal{O}$, attributes $\mathcal{A}$, and relations $\mathcal{R}$; triplets are $\langle s, r, o \rangle \in \mathcal{R}$. For the image $I$, let $\mathcal{B} = \{b_k\}_{k=1}^{R}$ denote sub-regions (boxes/masks) with associated labels/descriptions consistent with $\mathcal{G}^{\text{text}}$.

**Why not directly use Visual Genome (VG) Krishna et al. (2016)** VG offers valuable structure but has *limited scale*, annotation noise, and long-tail sparsity relative to modern pretraining needs; thus, it is insufficient alone for fine-tuning large VLMs. Our formulation treats scene-graph *annotations* as supervision units without requiring additional detectors, and is agnostic to the concrete source of those annotations.

## 3.3 NEGATIVES: ROLES AND PITFALLS (INTRA-IMAGE & CROSS-IMAGE)

Contrastive learning relies on negatives. Cross-image negatives are standard; however, *intra-image negatives* (mismatching a region with an unrelated text span from the *same* image) can be deceptively

hard and, if mishandled, push representations to ignore fine-grained correspondences. We therefore (i) explicitly control intra-image negatives and (ii) treat overlapping regions that refer to the *same* concept as *multi-positives* instead of false negatives.

## 3.4 Overlap in Scene Graphs: Multi-Positives by IoU

Scene graphs induce overlapping substructures (e.g., subject and subject-attribute regions). To our knowledge, we are the first to formalize *IoU-driven multi-positives* for region-text alignment: for a target region $b_i$ in image I ($b_i \in I$), the positive set is

$$\mathcal{P}(i) = \left\{ j \mid \text{IoU}(b_i, b_j) \geq \tau_{\text{pos}} \ \wedge \ b_j \in I \right\}, \tag{2}$$

and $\mathcal{N}(i)$ are the remaining candidates (cross-image) after removing $\mathcal{P}(i)$, with $\text{IoU}(b_i, b_j) \in [0, 1]$ and threshold $\tau_{\text{pos}} \in (0, 1)$. We additionally **mask intra-image negatives** with $\text{IoU}(b_i, b_j) < \tau_{\text{pos}}$ (i.e., same image but non-overlapping/weakly-overlapping regions), since they can be relevant to the *global* scene (coarse alignment) even if not to the target subregion, pushing them apart in a region-level loss would conflict with image-level alignment.

## 3.5 Region-Level Contrastive Alignment

To move beyond coarse image–text alignment, we introduce a *region-level contrastive loss* that aligns fine-grained sub-regions of an image with corresponding text spans (Fig. 3). Let $b_i$ denote a region proposal within image $I$, and let $\mathbf{v}_i^{\text{reg}}$ represent its embedding, obtained by pooling local features from the vision encoder over $b_i$. Similarly, let $\mathbf{t}_j^{\text{span}}$ denote the embedding of a span $d_j$ extracted from the scene graph. We define a *text span* as a variable-length segment of the caption associated with a sub-region of the image. Unlike prior work that uses only noun phrases or object labels, our spans may include longer descriptive expressions grounded in the visual scene.

For each region $b_i$, we define the set of positive spans $\mathcal{P}(i)$ as those that refer to the same object or attribute, including overlapping regions above the IoU threshold. Conversely, for each text span $\mathbf{t}_j^{\text{span}}$, we define its set of positive regions $\mathcal{P}(j)$ in the same manner.

The region-level contrastive loss is then defined as:

$$\begin{aligned}
\mathcal{L}_{\text{region}} = \ &- \frac{1}{R} \sum_{i=1}^{R} \log \frac{\sum_{j \in \mathcal{P}(i)} \exp\left(\text{sim}(\mathbf{v}_i^{\text{reg}}, \mathbf{t}_j^{\text{span}})/\tau\right)}{\sum_{k \in \mathcal{C}_t(i)} \exp\left(\text{sim}(\mathbf{v}_i^{\text{reg}}, \mathbf{t}_k^{\text{span}})/\tau\right)} \\
&- \frac{1}{M} \sum_{j=1}^{M} \log \frac{\sum_{i \in \mathcal{P}(j)} \exp\left(\text{sim}(\mathbf{t}_j^{\text{span}}, \mathbf{v}_i^{\text{reg}})/\tau\right)}{\sum_{k \in \mathcal{C}_v(j)} \exp\left(\text{sim}(\mathbf{t}_j^{\text{span}}, \mathbf{v}_k^{\text{reg}})/\tau\right)}
\end{aligned} \tag{3}$$

Here, $\mathcal{P}(i)$ and $\mathcal{P}(j)$ are multi-positive sets based on IoU and semantic unit overlap (see Sec. 3.4). The *masked candidate sets* in the denominator are defined as:

$$\mathcal{C}_t(i) = \left\{ k \mid \text{img}(\mathbf{t}_k) \neq \text{img}(\mathbf{v}_i) \right\} \cup \mathcal{P}(i), \qquad \mathcal{C}_v(j) = \left\{ k \mid \text{img}(\mathbf{v}_k) \neq \text{img}(\mathbf{t}_j) \right\} \cup \mathcal{P}(j).$$

Here $\text{img}(\cdot)$ returns the *image ID* associated with a given item: $\text{img}(\mathbf{v}_i^{\text{reg}})$ is the ID of the image containing region $i$, and $\text{img}(\mathbf{t}_k^{\text{span}})$ is the ID of the caption/span $k$ belongs to.

This formulation excludes the same image but non-positive region/span pairs from the denominator (i.e., those with IoU $< \tau_{\text{pos}}$ or mismatched semantic units), while still including cross-image candidates and in-image positives. This prevents the region-level loss from conflicting with global image–text alignment.

> *Why is the* log *outside the summation (second order summation)?* With multiple valid positives (e.g., overlapping regions or spans), the numerator aggregates their contributions. Placing the log outside evaluates the *total probability mass* assigned to all positives, so they reinforce rather than compete.

This design offers two key benefits. First, by treating IoU-overlapping regions as valid positives, the model avoids penalizing near-duplicate substructures that would otherwise be misclassified as

negatives. Second, aligning sub-regions with text spans explicitly enforces attribute–object binding at the *part level*, mitigating CLIP's bias towards large or central objects(Abbasi et al., 2025). In effect, $\mathcal{L}_{\text{region}}$ encourages attention to be distributed across finer details, improving compositional grounding.

### 3.6 RELATION AWARENESS: TRIPLET-LEVEL CONTRASTIVE LEARNING

Capturing relational structure remains a key challenge for vision–language models, as spatial and semantic relations are often underrepresented in web captions. Existing methods address this by (i) generating dense captions via language models or (ii) crafting scene-graph-aware negatives through rule-based perturbations (Huang et al., 2023). However, these approaches introduce noise or yield trivial, non-generalizable negatives. We propose a direct, *relation-aware* contrastive loss over object–relation–object triplets, explicitly extracted from captions and grounded in the image. This encourages models to learn *who does what to whom*, beyond global or region-level alignment.

**Triplet representations.** For a textual triplet $\langle s, r, o \rangle$, we construct a textual embedding $\mathbf{t}_{sro} = h_{\text{text}}(\langle s, r, o \rangle)$ (e.g., span-composed encoding of the subject, relation, and object). On the visual side, we use the complete image embedding $\mathbf{v} = f_\theta(I)$ rather than cropped regions, since the relation is defined at the scene level. Each image embedding is thus aligned against all triplets extracted from its caption.

**Setup.** Let $\mathcal{R} = \{(i, m)\}$ be all (image, triplet) pairs in the batch. Let $\mathcal{R}_i = \{(i, m) \in \mathcal{R}\}$ denote the triplets from image $i$. Let $N$ be the number of images.

$$
\mathcal{L}_{\text{relation}} = -\frac{1}{|\mathcal{R}|} \sum_{(i,m) \in \mathcal{R}} \log \frac{\exp\big(\text{sim}(\mathbf{v}^{(i)}, \mathbf{t}_m^{(i)})/\tau\big)}{\exp\big(\text{sim}(\mathbf{v}^{(i)}, \mathbf{t}_m^{(i)})/\tau\big) + \sum_{(j,n) \in \mathcal{R} \setminus \mathcal{R}_i} \exp\big(\text{sim}(\mathbf{v}^{(i)}, \mathbf{t}_n^{(j)})/\tau\big)}
$$
$$
- \frac{1}{|\mathcal{R}|} \sum_{(i,m) \in \mathcal{R}} \log \frac{\exp\big(\text{sim}(\mathbf{t}_m^{(i)}, \mathbf{v}^{(i)})/\tau\big)}{\exp\big(\text{sim}(\mathbf{t}_m^{(i)}, \mathbf{v}^{(i)})/\tau\big) + \sum_{\substack{j=1 \\ j \neq i}}^{N} \exp\big(\text{sim}(\mathbf{t}_m^{(i)}, \mathbf{v}^{(j)})/\tau\big)}. \tag{4}
$$

> *Why is the* log *inside the summation?* It makes each triplet a separate contrastive decision. With intra-image masking, every triplet scores against *cross-image* candidates only, so we average per-triplet log-likelihoods without penalizing same-image triplets.

This symmetric design ensures that (i) each image embedding retrieves its correct relational triplets (*image → triplet*), and (ii) each triplet embedding retrieves the image from which it was derived (*triplet → image*). By explicitly modeling triplets at the contrastive level, the model becomes sensitive to relational semantics such as subject–object role swaps and spatial ordering. Relation-level alignment with built-in sensitivity to role and word order; aggregating over all triplets per image drives joint relational modeling and a stronger compositional bias.

### 3.7 UNIFIED OBJECTIVE

Our final training criterion combines the three terms:

$$
\mathcal{L} = \lambda_{\text{region}} \mathcal{L}_{\text{region}} + \lambda_{\text{relation}} \mathcal{L}_{\text{relation}} + \lambda_{\text{global}} \mathcal{L}_{\text{global}}. \tag{5}
$$

The framework is *parameter-efficient*, plug-and-play over CLIP fine-tuning, and agnostic to the specific source of scene-graph annotations. All structured signals are consumed as supervision units for contrastive learning, yielding richer alignment without sacrificing zero-shot or retrieval performance.

## 4 EXPERIMENTS

### 4.1 DATASETS FOR FINE-TUNING

We finetune on the **Graph-Based Captioning (GBC)** (Hsieh et al., 2025) dataset, which provides *scene graphs* entities (boxes, attributes) and relational information unlike **CC3M/CC12M** that offer

a single caption per image and suffer an information imbalance (Abbasi et al., 2025). Triplets (subject–predicate–object) are extracted from these graphs and refined with **Qwen/LLM-based** parsing to mitigate noise and recover richer relations. We use the **GBC-1M** split, filtering for images with $\geq 5$ subregions and $\geq 10$ triplets, yielding **400k** training samples. The pipeline is modular: triplets can originate from dataset annotations or LLM-generated sources.

> *Triplet extraction.* We extract $\langle s, r, o \rangle$ triplets from captions using Qwen2.5-7B-Instruct. Prompting and implementation are in Appendix D.1. (Statistics in Appendix D.3.)

### 4.2 Implementation Details

We use **OpenCLIP** Ilharco et al. (2021) library **ViT-B/32** model, pretrained on **LAION-2B**. The model is fine-tuned with **LoRA adapters** over the filtered subset of GBC1M. Comprehensive details are mentioned in the Appendix D. We plan to release the code after acceptance.

### 4.3 Baselines and Evaluation Benchmarks

We compare our approach against several strong baselines. First, we include the standard **Pretrained CLIP** (ViT-B/32) Radford et al. (2021) evaluated in the zero-shot setting, as well as a **Fine-tuned CLIP** variant trained only on global image–text pairs without region or triplet level supervision. We consider **NegCLIP** from the ARO benchmark Yuksekgonul et al. (2023b), which fine-tunes CLIP with hard negative sampling. In addition, we reference **compositional baselines** reported in prior works, including results on SugarCrepe, CREPE, and What'sUp, where available.

We evaluate *compositional* reasoning beyond global alignment using three targeted benchmarks. **SugarCrepe** Hsieh et al. (2023) tests model sensitivity to object, attribute, and relation changes through controlled caption perturbations (*Replace*, *Swap*, *Add*) over fluent, plausible distractors. **Cola** Ray et al. (2023b) assesses attribute binding with objects while rejecting the distractors from the same part in some wrong configuration. **What'sUp** Kamath et al. (2023) is a tighlty controleld benchmark specifically focuses on spatial and relational reasoning. For broader multimodal representation analysis, we additionally evaluate on **UT-Zappos** Isola et al. (2015), which emphasizes fine-grained attribute understanding in product images. We follow standard protocol and report both **I2T** and **T2I** retrieval metrics to reflect fine-grained image–text alignment.

### 4.4 Compositional Evaluation

We evaluate on **SugarCrepe**, **Cola**, and **What'sUp**, which probe fine-grained compositional reasoning via controlled perturbations and relational tests. Tables 1, 2 and 3 compare our method against CLIP, NegCLIP, and recent baselines.

**SugarCrepe.** Table 1 shows consistent gains over CLIP, especially on *Add* (+16–25 pts) and *Swap* (+6–10 pts), with modest improvements on relation splits, indicating that structured supervision strengthens compositional understanding. *vs. NegCLIP:* While NegCLIP relies on perturbation-based hard negatives (replace/swap/add), SugarCrepe shows this alone does not ensure robust attribute–object binding. By incorporating structured information with global alignment, our approach delivers more reliable improvements under adversarially fluent negatives.

**Cola.** On the multi-object setting (Table 2), our method attains the highest performance across all objectives. These improvements indicate stronger role binding and more reliable reasoning over multiple objects, whereas CLIP, NegCLIP, and TripletCLIP often fail to disambiguate competing distractors. Our IoU-based region loss is particularly effective here, as it leverages sub-region annotations to capture fine-grained, cross-modal correspondences that enhance region-level depth understanding.

**What'sUp.** Table 3 shows our method achieves the best average accuracy, with clear gains on spatial/relational and multi-object splits. Subset A (selective relations) and Subset B (closed setup) present each image with four minimally different captions, making the task highly challenging; stronger scores here indicate learning of more generalizable relational concepts. Unlike COCO/GQA-spatial–tuned variants that benefit from distribution overlap, our model improves without dataset-specific supervision, reflecting better out-of-distribution generalization. Although

Table 1: **SugarCrepe results.** Accuracy (*Replace* (Obj/Att/Rel), *Swap* (Obj/Att), *Add* (Obj/Att). *Ours* achieves consistent improvements over CLIP across all categories. Bold indicates the best score and underline indicates the second-best.

| Methods | Replace | | | Swap | | Add | |
|---|---|---|---|---|---|---|---|
| | Object | Attribute | Relation | Object | Attribute | Object | Attribute |
| Human | 100 | 99 | 97 | 99 | 100 | 99 | 99 |
| CLIP | 90.92 | 80 | 69.13 | 61.2 | 64 | 77.16 | 68.2 |
| **Finetuned models** | | | | | | | |
| CLIP-FT | 90.92 | 79.69 | 64.01 | 60.82 | 64.26 | 84.67 | 78.76 |
| NegCLIP | 91.53 | 83.25 | 73.97 | **72.24** | 67.72 | 86.95 | 88.44 |
| CoN-CLIP | 93.58 | 80.96 | 63.30 | 59.18 | 65.16 | 87.29 | 79.62 |
| BLIP-SVGL | 53.69 | 52.41 | 47.43 | 44.89 | 56.00 | 45.87 | 50.57 |
| TSVLC (RB) | 91.34 | 81.34 | 64.15 | 68.16 | 69.07 | 79.49 | 91.33 |
| TSVLC (LLM+RB) | 88.13 | 76.78 | 62.73 | 64.08 | 66.67 | 75.80 | 81.07 |
| LaCLIP | 93.28 | 81.09 | 61.73 | 62.44 | 58.70 | 81.57 | 73.55 |
| CyCLIP | 80.87 | 66.12 | 57.54 | 53.88 | 52.11 | 71.48 | 65.75 |
| TripletCLIP | 94.43 | **85.53** | **80.94** | 69.80 | 69.82 | 90.40 | 86.27 |
| Cluster-Mask-CLIP | 86.13 | 75.13 | 64.65 | 66.67 | 63.36 | 74.92 | 71.24 |
| *RACA-CLIP (Ours)* | **94.73** | 85.15 | 72.11 | 71.00 | **70.50** | **93.40** | **93.06** |
| w.r.t CLIP | (+3.81) | (+5.15) | (+2.98) | (+9.80) | (+6.50) | (+16.24) | (+24.86) |

"Pairs" and "Set-of-4" accuracies (present in Appendix E.3) remain near chance for all models, this underscores the difficulty of resolving fine caption flips; we revisit this in the next section through intra-/inter-variance analysis.

## 4.5 ABLATION STUDIES

We conduct ablation experiments to understand the contribution of different components of our framework. Specifically, we analyze the role of loss terms, frozen encoders, learned representation structure, and weight interpolation in Appendix E.1.

**Weight Interpolation** We explore the trade-off between pretrained generalization and fine-tuned compositionality via:

$$W_{\text{interp}} = \alpha \, W_{\text{pretrained}} + \beta \, W_{\text{finetuned}}, \quad \alpha + \beta = 1.$$

Table 2: **Cola Multi-Object.** Joint accuracies (%, higher is better) for T2I, I2T, and GROUP (AND).

| Methods | Multi-Object (joint, %) | | |
|---|---|---|---|
| | T2I | I2T | GROUP |
| CLIP ViT-B/32 | 20.95 | 33.33 | 14.29 |
| NegCLIP | 16.19 | 32.38 | 11.43 |
| TripletCLIP | 10.95 | 32.38 | 9.05 |
| *RACA-CLIP (Ours)* | **31.43** | **41.43** | **19.05** |

Table 4 reports the best ImageNet-1K zero-shot accuracy and COCO/Flickr30k R@1. Unlike Recall@5 where NegCLIP and TripletCLIP benefit from hard negatives R@1 is more demanding, as it requires truly capturing subtle cross-modal cues beyond bag-of-words. We address this by weighted interpolation, injecting compositional penalties into standard embeddings to make representations more sensitive. While pure fine-tuning boosts compositionality at the cost of zero-shot, interpolation ($\beta > 0$) balances both, improving alignment without sacrificing generalization.

## 4.6 REPRESENTATION ANALYSIS

We evaluate our method with UT-Zappos-50k, a fine-grained benchmark where attributes differ subtly. We measure mean cosine similarities for *intra/cross*-class pairs within each modality and inspect per-modality variances (present in Appendix Table 13). This probes object attribute disentanglement, where VLMs often conflate compositional elements. Our method achieves high performance in terms of mean accuracy over correct matches (cross-modality retrieval). Compared to baselines (Table 5), our representations exhibit (i) lower cross-class similarities in both modalities (shown with ↓), indicating better class separability and reduced modality-specific hubness; (ii) higher intra-class cohesion, especially on the text side; and (iii) more balanced variance (presented in Appendix Table 13) across object and attribute dimensions, while text-attribute variance remains strongly compressed across all models, hinting at a **potential bottleneck for attribute grounding**.

**Object–Attribute Bias.** Objects remain easier to align than attributes, reflecting object-centric biases in VLMs Schrodi et al. (2025). This stems from annotation imbalance, as attributes rarely appear independently and are tied to specific objects, making disentanglement harder. Simply adding hard negatives cannot fix this, as models tend to bias toward dominant features unless massively oversampled. In contrast, leveraging scene graphs probes object–attribute dynamics directly, injecting structured variation that helps VLMs extrapolate beyond surface correlations and achieve stronger disentanglement verified in Table 5. Relative Modality Gap is discussed in the Appendix.

Table 3: **What'sUp.** Spatial/relational reasoning with controlled caption flips (e.g., above/below). Scores are accuracies (%).

| Model | What'sUp | | COCO-spatial | | GQA-spatial | | Avg. |
|---|---|---|---|---|---|---|---|
| | Subset A | Subset B | One-obj. | Two-obj. | One-obj. | Two-obj. | |
| CLIP ViT-B/32 | 30.3 | 31.6 | 43.7 | 51.1 | 46.5 | 47.4 | 41.8 |
| FT on train COCO/GQA-spatial | 28.2 | 25.2 | **67.2** | **60.7** | **64.4** | **54.6** | **50.0** |
| FT on LAION-4M-prep | 31.6 | 34.6 | 43.1 | 48.9 | 44.3 | 50.9 | 42.2 |
| FT on LAION-4M-prep + neg. cap. | 32.0 | 26.5 | 39.9 | 48.9 | 47.3 | 45.7 | 40.1 |
| NegCLIP | 32.5 | 36.3 | 47.4 | 46.4 | 45.3 | 46.7 | 42.4 |
| TripletCLIP | 26.9 | 32.6 | 46.5 | 51.5 | 56.65 | 32.0 | 41.03 |
| *RACA-CLIP (Ours)* | **34.46** | **37.50** | 50.24 | 55.23 | 55.66 | 51.19 | 47.4 |
| Random chance | 25.0 | 25.0 | 50.0 | 50.0 | 50.0 | 50.0 | 41.7 |

Table 4: **Comparison of CLIP on Zero-shot Classification and Retrieval.** Below, we report ImageNet-1K zero-shot classification accuracy and retrieval performance (Recall@1).

| Methods | Zero-shot Classification | | Retrieval (R@1) | | | |
|---|---|---|---|---|---|---|
| | ImageNet1K | | Image-to-Text | | Text-to-Image | |
| | top-1 | top-5 | MSCOCO | Flickr30k | MSCOCO | Flickr30k |
| CLIP | 55.96 | 83.41 | 50.02 | 78.59 | 30.35 | 59.72 |
| NegCLIP | 48.92 | 77.93 | 56.84 | 83.03 | **41.56** | 68.73 |
| Triplet CLIP | 41.10 | 70.47 | 31.30 | 59.66 | 28.89 | 57.08 |
| *RACA-CLIP (Ours)* | **58.11** | **83.67** | **57.54** | **85.50** | 39.36 | **69.27** |

Table 5: **Multimodal analysis on UT-Zappos-50k.** Lower *Cross* and moderate intra-class separation indicate less hubness and better alignment. **RACA-CLIP (Ours)** shows the best overall accuracy (T2I and I2T) and lowest cross-class similarity in both object and attribute groups.

| Method | Acc (%) | Image (intra) | | Image (cross) | | Text (intra) | | Text (cross) | |
|---|---|---|---|---|---|---|---|---|---|
| | | Obj | Attr | Obj↓ | Attr↓ | Obj | Attr | Obj↓ | Attr↓ |
| TripletCLIP | 31.51 | 0.871 | 0.889 | 0.849 | 0.854 | 0.955 | 1.000 | 0.884 | 0.903 |
| NegCLIP | 34.40 | 0.816 | 0.843 | 0.785 | 0.792 | 0.927 | 1.000 | 0.833 | 0.856 |
| *RACA-CLIP (Ours)* | **38.21** | 0.763 | 0.794 | **0.716** | **0.728** | 0.885 | 1.000 | **0.783** | **0.803** |

## 4.7 FINE-GRAINED UNDERSTANDING

**How well does RACA-CLIP handle fine-grained compositional understanding?** We further evaluate compositional representations on **MMVP-VLM** (Tong et al. (2024)), a multiple-choice VQA-style benchmark over natural images with visual attribute categories (orientation, specific features, state, quantity, position/relations, color, structure, **text**, viewpoint). As shown in Table 6, RACA-CLIP achieves the highest average accuracy (33.3%), improving over vanilla CLIP (27.4%) by 5.9 absolute points (∼21% relative) while also outperforming NegCLIP (Yuksekgonul et al. (2023a)), TripletCLIP (Patel et al. (2024)), and CE-CLIP (Zhang et al. (2024)). Importantly, this gain is obtained **without any explicit hard-negative mining**, suggesting that region- and triplet-aware positive supervision is already sufficient to induce stronger fine-grained compositional structure.

Table 6: **MMVP-VLM:** Compositional accuracy (%) across visual patterns. Symbols denote visual attributes (following the MMVP-VLM Tong et al. (2024) taxonomy): 🧭 Orientation and Direction, 🔍 Presence of Specific Features, 🔄 State and Condition, 🔢 Quantity and Count, 🎯 Positional and Relational Context, 🎨 Color and Appearance, ⚙️ Structural Characteristics, **A** Texts, 📷 Viewpoint and Perspective. Bold indicates the best score and underline indicates the second-best.

| Method | Backbone | Res | 🧭 | 🔍 | 🔄 | 🔢 | 🎯 | 🎨 | ⚙️ | A | 📷 | Avg. |
|---|---|---|---|---|---|---|---|---|---|---|---|---|
| CLIP | ViT-B/32 | 224 | 26.7 | 6.7 | 40.0 | 6.7 | **26.7** | 66.7 | 33.3 | **26.7** | 13.3 | 27.4 |
| NegCLIP | ViT-B/32 | 224 | 20.0 | 13.3 | 46.7 | 13.3 | 20.0 | 46.7 | 20.0 | 26.7 | 26.7 | 25.9 |
| TripletCLIP | ViT-B/32 | 224 | 6.7 | 13.3 | 40.0 | 6.7 | 13.3 | 53.3 | 13.3 | 13.3 | 20.0 | 20.0 |
| CE-CLIP | ViT-B/32 | 224 | 6.7 | 20.0 | 46.7 | **13.3** | 20.0 | 53.0 | 20.0 | 20.0 | 13.3 | 23.7 |
| *RACA-CLIP (Ours)* | ViT-B/32 | 224 | **26.7** | **20.0** | **60.0** | 6.7 | 20.0 | **80.0** | **40.0** | 13.3 | **33.3** | **33.3** |

Table 7: **Robustness to scene-graph noise.** We randomly corrupt $k\%$ of region spans and triplet relations in the input scene graphs during training of RACA-CLIP and report average accuracies (%) over compositional benchmarks and the fine-grained MMVP-VLM Tong et al. (2024) diagnostic. CLIP does not consume scene graphs and is therefore unaffected by this perturbation.

| Model | WhatsUp | COLA | Wino-ground | Sugar-Crepe | SugarCrepe++ | MMVP-VLM | Avg. |
|---|---|---|---|---|---|---|---|
| CLIP | 41.8 | 22.8 | 16.4 | 72.9 | 54.6 | 27.4 | 39.3 |
| *RACA-CLIP (0% noise)* | **47.0** | **30.6** | **18.3** | **83.0** | **65.1** | **33.3** | **46.2** |
| *RACA-CLIP (10% noise)* | 45.9 | 30.0 | 17.8 | 82.93 | 62.90 | 29.0 | 44.8 |
| *RACA-CLIP (20% noise)* | 45.2 | 29.1 | 16.6 | 81.32 | 61.35 | 28.15 | 43.6 |

## 4.8 ROBUSTNESS ANALYSIS

**How sensitive is RACA-CLIP to noisy scene-graph supervision?** Since RACA-CLIP relies on region spans and relational triplets extracted from scene-graph annotations, we probe its dependence on annotation quality by injecting synthetic noise into the training graphs: we randomly corrupt $k\%$ of region spans and triplet relations while keeping all other settings fixed. Table 7 summarizes performance across all compositional benchmarks. RACA-CLIP reaches an average accuracy of 46.2% with clean graphs, compared to 39.3% for CLIP. With 10% and 20% corrupted supervision, the averages decline smoothly to 44.8% and 43.6% yet remain consistently above CLIP. The degradation is gradual rather than abrupt across WhatsUp (Kamath et al. (2023)), COLA (Ray et al. (2023b)), Winoground (Thrush et al. (2022)), SugarCrepe (Hsieh et al. (2023)), SugarCrepe++ (Dumpala et al. (2024)), and MMVP-VLM (Tong et al. (2024)). These results show that while RACA-CLIP benefits from higher-fidelity graphs, it is not brittle: its compositional gains persist under moderate noise, indicating that the structured signals function as a soft inductive bias rather than requiring perfectly accurate scene graphs.

## 4.9 OTHER ANALYSES

Further analyses—including Winoground, SugarCrepe++, causal/probing, backbone scaling, and our idea of not using hard-negatives, discussion—are provided in continuation of Appendix E.5.

## 5 CONCLUSION

We proposed a parameter-efficient framework, dubbed as RACA-CLIP, that injects scene-graph inductive bias into CLIP training via three complementary objectives, without explicitly relying on hard negatives to promote compositional awareness. RACA-CLIP method yields consistent gains across diverse benchmarks, particularly in relation and multi-object settings, while maintaining or improving zero-shot performance. Representation and frozen model analyses reveal deeper insights into modality structure, showing more disentangled, stable spaces with reduced hubness. A key limitation is reliance on the quality and coverage of scene-graph signals, which may impact performance. Future work may explore scaling structured supervision and enhancing graph extraction for broader generalization, as well as exploring the applicability of RACA-CLIP to diffusion models.

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

APPENDIX

We provide extended literature, additional results, implementation details, and extended analyses in the following appendix.

## A    USE OF LARGE LANGUAGE MODELS (LLMs)

We leveraged LLMs for refining, grammar correction, and brainstorming to create visually appealing charts/figures.

## B    RELATED WORK

### B.1    CONTRASTIVE PRETRAINING FOR VISION–LANGUAGE ALIGNMENT

Contrastive pretraining has become the dominant paradigm for vision–language alignment, with models such as CLIP (Radford et al., 2021) and ALIGN (Jia et al., 2021) training dual encoders on hundreds of millions of web-scraped image–text pairs to maximize similarity between paired inputs while minimizing similarity with negatives. This simple yet scalable objective has produced universal cross-modal representations with strong zero-shot transfer across tasks including image classification, retrieval, captioning, and even as backbones for large multimodal and generative models (Mokady et al., 2021; Li et al., 2022a; Alayrac et al., 2022). The effectiveness of this paradigm is closely tied to scale: massive datasets such as LAION-400M/5B (Schuhmann et al., 2022) compensate for noise, enabling models to learn robust global alignments. However, several studies reveal that the contrastive objective often encourages shortcut learning—successfully separating easy negatives while failing on fine-grained distinctions of attributes, relations, or word order—leading to a "bag-of-objects" behavior rather than structured compositional understanding (Thrush et al., 2022; Yuksekgonul et al., 2023b). To overcome these issues, extensions have explored filtering or augmenting data (Wang et al., 2024; Sahin et al., 2023b), generating synthetic captions (Patel et al., 2024; Doveh et al., 2023), or integrating auxiliary objectives such as masked modeling and image–text matching (Sinha et al., 2021; Li et al., 2023), while methods like CyCLIP (Goel et al., 2022) incorporate geometric constraints to stabilize embeddings. Yet, despite these efforts, the fundamental limitation remains: contrastive functions defined solely over global embeddings lack the inductive bias required for compositional reasoning, unlike our approach, where we inject structured representations directly into the contrastive framework.

### B.2    LIMITATIONS OF COMPOSITIONALITY IN VLMS

Benchmarks such as Winoground (Thrush et al., 2022), ARO (Yuksekgonul et al., 2023b), CREPE (Ma et al., 2023), What's "up" (Kamath et al., 2023), Cola Ray et al. (2023b) and SugarCrepe (Hsieh et al., 2023) repeatedly show that state-of-the-art models perform near chance on tasks requiring systematic generalization, spatial reasoning (Kamath et al., 2023), or fine-grained linguistic distinctions (Yuksekgonul et al., 2023b). These failures reveal a "bag-of-objects/words" bias, where models succeed at coarse alignment but ignore structured bindings, undermining their semantic understanding. Attempts to improve compositionality via rule-based negatives (Wang et al., 2024), LLM-generated captions (Sahin et al., 2023b), or synthetic data (Wang et al., 2024) yield partial gains but remain limited by data quality, linguistic unnaturalness, or domain gaps, Unlike prior work, we use scene-graph annotations (even at limited scale) to enable high-quality, fine-grained learning in VLMs.

### B.3    HARD NEGATIVES AND THEIR TRADE-OFFS

Hard negatives—minimally perturbed captions that alter attributes, relations, or word order—are widely used to improve compositionality (Yuksekgonul et al., 2023b; Wang et al., 2024; Rösch et al., 2024). They can be generated via rule-based swaps (e.g., NegCLIP), scene graphs (Huang et al., 2023; Herzig et al., 2023), or LLMs (Sahin et al., 2023b), and typically raise fine-grained benchmark scores. However, applying these objectives at the *global* embedding level can destabilize alignment—making models brittle to meaning-preserving edits and hurting zero-shot recognition and retrieval (Kamath et al., 2024). Prior work also targets negation and unimodal composition

(Alhamoud et al., 2025; Koishigarina et al., 2025). In contrast, we go beyond attribute–object pairs to explicitly model *relations*, using structured, part-aware objectives that strengthen cross-modal alignment and yield more reliable reasoning.

### B.4 STRUCTURED REPRESENTATIONS: SCENE GRAPHS FOR FINE-GRAINED ALIGNMENT

Structured representations such as scene graphs provide a natural way to model objects, attributes, and relations, and have been applied to captioning, retrieval, and image generation (Hsieh et al., 2025). In vision–language modeling, contrastive objectives remain the most effective pretraining paradigm (Zeng et al., 2022; Kamath et al., 2023; Zhao et al., 2023), but standard CLIP-style training relies on *global* image–text embeddings, which tends to discard fine-grained structure and encourage "bag-of-objects" behavior. We instead leverage scene-graph annotations to modify the learning objective without changing the CLIP architecture: (i) a *region-level* IoU-weighted multi-positive loss aligns image sub-regions with their corresponding textual spans, and (ii) a *relation-aware* loss aligns image features with $\langle s,r,o \rangle$ triplets extracted from captions, strengthening attribute binding and relation understanding while preserving global alignment. RACA-CLIP is related to prior structure-aware methods such as Structure-CLIP( Huang et al. (2023)) and SGVL( Herzig et al. (2023)), but incorporates structure in a lighter-weight way. Structure-CLIP constructs text-side scene graphs to generate hard negative captions and adds a Knowledge-Enhanced text encoder, injecting structure only on the language side and increasing architectural complexity; SGVL attaches scene-graph prediction heads and additional graph tokens to the vision backbone and also relies on graph-based hard negatives. In contrast, RACA-CLIP keeps the CLIP dual encoder unchanged and uses GBC-derived region–span pairs and triplets solely to define *positive* region- and relation-aware contrastive losses in the shared embedding space. Scene-graph machinery is used only during training, and inference requires only raw images and text, making RACA-CLIP a drop-in, structure-enhanced replacement for CLIP. RACA-CLIP uses *no* hard negatives: scene graphs only define *positive* region–span and $\langle s,r,o \rangle$ triplet alignments, while negatives remain the standard in-batch pairs as in vanilla CLIP.

## C   IoU-WEIGHTED GLOBAL CONTRASTIVE LOSS

To balance fine-grained objectives with stable global alignment, we reweight negatives in the global InfoNCE loss by structure-derived similarity. Specifically, $w_{ij} \in (0,1]$ down-weights negatives whose paired text/image share high structural overlap with the anchor (i.e., semantically adjacent but not exact matches). Formally,

$$w_{ij} = \exp\big( -\alpha \cdot \Delta_{\text{IoU}}(i,j)\big), \quad \Delta_{\text{IoU}}(i,j) = 1 - \overline{\text{IoU}}(\mathcal{B}_i, \mathcal{B}_j), \tag{3}$$

where $\overline{\text{IoU}}$ aggregates region overlaps induced by aligned scene-graph units, and $\alpha > 0$ controls the sharpness of weighting. The resulting loss is

$$\mathcal{L}_{\text{IoU-global}} = -\frac{1}{N} \sum_{i=1}^{N} \log \frac{\exp\big(\text{sim}(\mathbf{v}_i, \mathbf{t}_i)/\tau\big)}{\sum_{j=1}^{N} w_{ij} \exp\big(\text{sim}(\mathbf{v}_i, \mathbf{t}_j)/\tau\big)} + \text{sym. (text} \rightarrow \text{image)}. \tag{4}$$

This formulation preserves global semantics while preventing the model from over-penalizing near-positive structures. In practice, however, adding this weighting scheme to the global contrastive objective did not yield improvements on compositional benchmarks or zero-shot classification. We attribute this to dataset characteristics, though the approach may prove more beneficial on noisier or less curated graph-based datasets.

## D   IMPLEMENTATION DETAILS

We use **ViT-B/32**, pretrained on **LAION-2B**, implemented with the **OpenCLIP** Ilharco et al. (2021) library. The model is fine-tuned for **26 epochs** using **AdamW** (lr=$1 \times 10^{-5}$, weight decay=0.01) with $500$ warm-up steps followed by cosine decay. Training is performed in **bfloat16** precision with **DDP** via HuggingFace `accelerate` Gugger et al. (2022), using a batch size of **120** per GPU. For efficient adaptation, we apply **LoRA adapters** (rank=16, $\alpha = 32$, dropout=0.1) to both text and vision transformers. Our objective combines a global contrastive loss, a region-level IoU-masked contrastive loss, and a triplet contrastive loss. All experiments are run on **NVIDIA Tesla V100 GPUs**.

**Practical Overhead vs. Plain CLIP-FT.** RACA-CLIP fine-tuning incurs only a *modest* overhead relative to plain CLIP fine-tuning. The additional cost arises from (i) region–span similarities with IoU-masked multi-positives and (ii) image–triplet similarities, which scale with the (sparse) number of regions and triplets per image. These introduce extra matrix multiplications but no new parameters beyond the LoRA adapters. On V100 GPUs, training with the IoU-weighted global contrastive variant required approximately **7 GPU-hours**, with wall-clock and memory slightly higher than CLIP-FT yet still lower than NegCLIP pipelines. Unlike our approach, NegCLIP and TripletCLIP rely on LLMs to generate synthetic negatives—either images or captions—which adds a separate preprocessing stage and increases total training cost. In contrast, our pipeline is self-contained and does not require auxiliary generation, making it both computationally lighter and easier to scale.

## D.1 PROMPTING DETAILS FOR TRIPLET EXTRACTION

We used a prompting strategy with Qwen2.5-7B-Instruct to extract subject–relation–object triplets from captions. The instruction given to the LLM was:

**Motivation.** Before finalizing the pipeline, we experimented with several prompting styles for dataset adaptation, including zero-shot extraction, chain-of-thought prompting, and different formatting constraints. Among these, the strategy described below consistently yielded the most accurate and structured triplets, balancing coverage with low noise.

**Why Qwen2.5-7B-Instruct?** We selected Qwen2.5-7B-Instruct for its balance of accuracy, efficiency, and open availability. Compared to larger LLMs, it provides strong instruction-following capability while being lightweight enough for multi-GPU parallelization across millions of captions. Qwen models have been optimized for multilingual and structured extraction tasks, making them well-suited for relation parsing without requiring additional fine-tuning. This allowed us to reliably obtain consistent triplets at scale with manageable computational cost.

---

**Prompting Strategy**

```
You are an information extractor for long, detailed captions.  From
the given text, extract atomic relational triplets in the EXACT format
<subject , relation , object>, one per line.

Rules:

1) Emit ONLY triplet lines (no numbering/explanations).

2) Split conjunctions into separate atomic triplets; keep each triplet
minimal.

3) Resolve obvious pronouns to antecedents when unambiguous.

4) Use verbs or short verb phrases for relations; prefer canonical
forms.

5) Use has-attribute for colors/materials/states.

6) Spatial relations:  on, in, under, above, below, behind, in front
of, next to, near, between, across, through, around, toward, beside,
across from.

7) Structural relations:  part-of, made-of, attached to, connected
to, supported by, covered by, surrounded by, enclosed by, filled with,
contain(s).

8) Interaction/action:  hold, wear, carry, ride, drive, eat, drink,
play, look at, talk to, chase, follow, sit on, stand on, hang from,
open, close.

9) Temporal/weather:  e.g., time , is , <value>; weather , is ,
<value>.

10) Quantity/comparison:  has-quantity, taller than, bigger than,
smaller than.

11) Ownership/identity/purpose/causality:  belong to, owned by, is-a,
located in, used for, caused by.

12) Max 12 triplets; prioritize salient, non-redundant relations.
```

---

**Few-Shot Examples**

```
Below we show representative input captions and the corresponding
extracted triplets used as few-shot demonstrations.
```

**Caption:** ``A hiker wearing a yellow raincoat carries a backpack and holds a walking stick.''

**Triplets:**

```
<hiker , wear , yellow raincoat>
<hiker , carry , backpack>
<hiker , hold , walking stick>
```

**Caption:** ``A storefront with bright neon signs has its glass door closed; the interior lights are on and the floor is wet.''

**Triplets:**

```
<signs , has-attribute , neon>
<storefront , has-attribute , bright>
<door , has-attribute , glass>
<door , has-attribute , closed>
<lights , has-attribute , on>
<floor , has-attribute , wet>
```

**Caption:** ``A steel bridge with wooden planks is supported by concrete pillars; cables are attached to the towers and connected to the deck, which is covered by a thin layer of snow.''

**Triplets:**

```
<bridge , made-of , steel>
<planks , made-of , wood>
<planks , part-of , bridge>
<bridge , supported by , pillars>
<pillars , made-of , concrete>
<cables , attached to , towers>
<cables , connected to , deck>
<deck , covered by , snow>
```

## D.2 RELATION FREQUENCY VS. WHAT'SUP BENCHMARK RELATIONS

To better understand the gap in compositional reasoning performance, we compare the frequency of relations used in the What'sUp benchmark (focused on spatial relations) against our generated triplets. Table 8 shows that these relations occur far less frequently in our dataset than the dominant relations reported in Table 9.

Table 8: **Frequencies of What'sUp benchmark relations in our extracted triplets.** Counts are raw frequencies before canonicalization.

| Relation | Count | Relation | Count |
|----------|-------|----------|-------|
| on | 328,303 | behind | 11,184 |
| in | 116,906 | next to | 13,391 |
| left | 3,783 | beneath | 1,621 |
| right | 3,868 | above | 41,363 |
| top | 2,350 | | |

We observe that while relations such as *has-attribute*, *wear*, or *include* dominate the extracted triplets (hundreds of thousands of occurrences), spatial relations central to What'sUp appear with much lower frequency (often by one to two orders of magnitude). This imbalance suggests that our dataset needed to have better diversity in spatial relations, which directly limits performance on set and pair

accuracy metrics in What'sUp, where distinguishing between fine-grained spatial configurations is critical.

## D.3 RELATION DIVERSITY IN EXTRACTED TRIPLETS

Our prompting pipeline produced a wide variety of relations. Table 9 reports the 50 most frequent relations (excluding *has-attribute*) out of the full set of 89,241 unique relation strings (before canonicalization). The long-tailed distribution indicates that while a small number of relations dominate, there is substantial coverage across diverse relation types.

Table 9: **Top-50 most frequent relations in extracted triplets (excluding *has-attribute*).** Counts are raw frequencies before canonicalization.

| Relation | Count | Relation | Count | Relation | Count |
|---|---|---|---|---|---|
| suggest | 439,826 | part-of | 50,749 | on left | 28,469 |
| on | 328,303 | filled with | 50,662 | on right | 27,626 |
| wear | 210,261 | adorned with | 46,380 | create | 27,467 |
| include | 178,263 | dress in | 44,154 | position | 26,936 |
| is | 175,521 | stand on | 44,115 | for | 26,926 |
| made-of | 169,732 | possibly | 42,785 | appear to be | 26,559 |
| has | 134,608 | above | 41,363 | engage in | 23,694 |
| have | 118,770 | indicate | 40,356 | consist of | 23,393 |
| in | 116,906 | has-pattern | 37,844 | from | 22,951 |
| feature | 116,290 | add | 34,686 | appear | 22,768 |
| has-quantity | 106,037 | display | 34,576 | contrast with | 20,942 |
| convey | 89,885 | at | 32,946 | contain | 20,839 |
| hold | 83,160 | adorn | 32,123 | has-design | 20,657 |
| has-color | 79,943 | look at | 32,028 | reflect | 20,499 |
| with | 76,013 | focus on | 31,623 | provide | 20,409 |

## D.4 STRUCTURED SUPERVISION AND COMPUTATIONAL OVERHEAD

**Scene-graph preprocessing.** RACA-CLIP does not require any additional scene-graph extraction beyond the region-level annotations already available in GBC. The only extra preprocessing step is LLM-based triplet extraction, performed once offline over the training captions (approximately 3 hours on 6×V100 GPUs). After this one-time step, training proceeds identically to standard CLIP fine-tuning, with scene-graph regions and triplets entering only through the loss functions.

**Training and inference cost.** The model retains the same ViT-B/32 and Transformer text encoder as CLIP, with no new modules or architectural changes. During fine-tuning, global images, region crops, captions, region descriptions, and triplet texts are batched into a single `encode_image` and `encode_text` call, so the backbone FLOPs per forward remain unchanged; the added cost stems primarily from processing multiple views and computing additional structured losses. On GBC-1M with 6×V100 GPUs, this results in roughly a 3× increase in wall-clock time compared to a CLIP-only baseline on the same data. At inference, RACA-CLIP relies solely on the global image–text pair, yielding the same parameter count, FLOPs, and latency as CLIP with no additional overhead.

# E  ABALATION

## E.1  SENSITIVITY TO LOSS WEIGHTS

We vary the relative weighting ($\alpha$, $\beta$, $\gamma$) of the three loss terms to test robustness to hyperparameter choices.

Table 10: **Sensitivity to loss weights.** Results showing performance (accuracy %) under different weight settings of global ($\alpha$), region ($\beta$), and triplet ($\gamma$) losses.

| $\alpha$ | $\beta$ | $\gamma$ | Accuracy on Sugar Crepe(%) | Notes |
|------|------|------|------|------|
| 1.0 | 0.5 | 0.5 | 81.6 | Balanced, baseline |
| 1.0 | 1.0 | 0.5 | 82.6 | Higher region weight |
| 1.0 | 0.5 | 1.0 | 79.6 | Higher triplet weight |
| 0.5 | 1.0 | 1.0 | 77.9 | Lower global weight |
| 1.5 | 0.5 | 0.5 | 75.8 | Overweighted global |
| 1.0 | 1.0 | 1.0 | 80.02 | All equal |

## E.2  FROZEN VS. UNFROZEN ENCODERS

We conduct an ablation comparing different freezing strategies during fine-tuning. Specifically, we consider (i) freezing the text encoder and training the vision branch, and (ii) freezing the vision encoder while training the text side. Table 11 reports results across retrieval and compositional benchmarks.

Table 11: **Frozen encoder ablation.** Retrieval performance is reported as Recall@1 (%) for Flickr30k and COCO; compositional performance is measured on What'sUp and SugarCrepe.

| Task / Benchmark | Text Frozen (train vision) | Vision Frozen (train text) |
|------|------|------|
| Flickr30k I2T | 84.81 | 77.00 |
| Flickr30k T2I | 69.21 | 66.00 |
| COCO I2T | 59.02 | 48.50 |
| COCO T2I | 39.52 | 36.95 |
| VGQA (one-obj) | 46.66 | 56.40 |
| VGQA (two-obj) | 52.31 | 42.44 |
| Controlled Set A | 29.85 | 30.10 |
| Controlled Set B | 29.66 | 32.60 |
| SugarCrepe (avg.) | 77.32 | 81.13 |

We observe two clear regimes: (i) For **retrieval tasks** (Flickr30k, COCO), freezing the text encoder and updating the vision side yields stronger performance. This suggests that pretrained vision features benefit from task-specific adaptation to improve instance-level matching. (ii) In contrast, for **compositional benchmarks** (SugarCrepe, What'sUp), freezing the vision encoder and training the text side produces consistently higher accuracy. This indicates that while visual features already encode rich object, attribute, and relational cues, the multimodal text embeddings require additional supervision to avoid collapse and to better capture fine-grained compositional structure.

These findings connect directly with our representation analysis (Section 4.6), where we observed collapsed distributions in attribute-related text embeddings. Updating the text encoder helps counteract this collapse, ensuring that the contrastive function aligns attributes and relations more meaningfully. Together, this ablation highlights that **retrieval and compositional reasoning stress different parts of the model**: vision-side adaptation supports global retrieval alignment, while text-side adaptation is crucial for structured compositional reasoning.

### E.3 RELATIVE MODALITY GAP

To better understand the alignment between visual and textual modalities, we measure the *relative modality gap* the difference in intra and inter-modal similarity distributions.

We quantitatively assess the alignment between modalities through the *Relative Modality Gap* (RMG), defined as the discrepancy between intra-modal and cross-modal similarities. Table 12 summarizes RMG-related metrics across different training setups.

- **Triplet CLIP**: Achieves an RMG of approximately 0.73, with strong intra-modal alignment (image-image cosine: 0.61, text-text cosine: 0.84) but relatively weaker cross-modal similarity (match cosine: 0.26). This suggests a modality imbalance that may hinder compositional reasoning.

- **Negative-CLIP Baseline**: Shows slightly improved cross-modal similarity (0.28), but significantly lower intra-modal cohesion (both image-image and text-text around 0.50), leading to an RMG of 0.59. The drop in intra-modal structure may reduce robustness in fine-grained composition tasks.

- **Ours (Epoch 15)**: Achieves the best balance with a reduced RMG of 0.61 and improved cross-modal match (0.37). Text-text cohesion remains strong (0.77), and image-image alignment is more moderate (0.43), indicating a better-integrated multimodal space.

These findings suggest that narrowing the modality gap not simply maximizing individual modality coherence can improve compositional generalization. A more balanced embedding space supports flexible recombination of concepts across modalities, which is essential for structured reasoning tasks.

**Conclusion:** Lower relative modality gap correlates with more disentangled, compositionally aware representations. This highlights the importance of structural objectives that go beyond simple instance matching and instead promote modality-aware integration.

| Model | Match Cos | Img-Img Cos | Txt-Txt Cos | RMG |
|---|---|---|---|---|
| Triplet Model | 0.2598 | 0.6128 | 0.8444 | 0.7317 |
| Neg-CLIP | 0.2847 | 0.5021 | 0.5092 | 0.5913 |
| *RACA-CLIP (Ours)* | 0.3690 | 0.4296 | 0.7703 | 0.6120 |

Table 12: Relative Modality Gap (RMG) and similarity scores across models.

Table 13: **Intra variance in image and text embeddings.** Higher variance implies better distributed representations and less collapse. **RACA-CLIP (Ours)** achieves the strongest intra-class diversity across both modalities. Note that the entries of the table (variances) need to be multiplied by $10^{-6}$.

| Method | Img-Obj | Img-Attr | Txt-Obj | Txt-Attr |
|---|---|---|---|---|
| TripletCLIP | 23.07 | 19.45 | 4.43 | 0.00 |
| NegCLIP | 31.27 | 26.95 | 6.86 | 0.00 |
| *RACA-CLIP (Ours)* | **41.25** | **33.98** | **9.85** | 0.00 |

### E.4 SCORES ON WHAT'SUP BENCHMARK

The table above reports comprehensive results on the What'sUp benchmark. We note that CLIP fine-tuned on COCO and GQA-Spatial achieves higher scores, largely because these datasets overlap with the evaluation distribution, giving that variant an advantage. By contrast, our model is not fine-tuned on either COCO or GQA, making the evaluation more fair but naturally yielding lower scores. Specifically, our set and pair accuracies are reported as zero consistent with prior works though in some runs we observed non-zero scores. However, these came at the expense of reduced individual accuracy, so we report the more stable runs. Importantly, What'sUp explicitly forces the model to resolve relational perturbations, making relation diversity a key factor. Standard datasets rarely

Table 14: **What'sUp** spatial/relational evaluation with caption flips (e.g., *above ↔ below*). We report accuracies (%) on Subsets A/B and transfer splits COCO-spatial and GQA-spatial (one/two objects), plus the overall average. *RACA-CLIP (Ours)* attains the best overall (**47.4**); the COCO/GQA-tuned CLIP variant leads only on its matched splits. Best/second-best are **bold**/underlined.

| Model | What'sUp Subset A | | | What'sUp Subset B | | | COCO-spatial | | GQA-spatial | | Indiv. Avg. |
|---|---|---|---|---|---|---|---|---|---|---|---|
| | Indiv. | Pairs | Set of 4 | Indiv. | Pairs | Set of 4 | One-obj. | Two-obj. | One-obj. | Two-obj. | |
| CLIP ViT-B/32 | 30.3 | 0.5 | 0.0 | 31.6 | 1.0 | 0.0 | 43.7 | 51.1 | 46.5 | 47.4 | 41.8 |
| FT on train COCO-spatial, GQA-spatial | 28.2 | 7.3 | 0.0 | 25.2 | 2.0 | 0.0 | 67.2 | 60.7 | 64.4 | 54.6 | 50.0 |
| FT on LAION-4M-prep | 31.6 | 1.0 | 0.0 | 34.6 | 2.9 | 0.0 | 43.1 | 48.9 | 44.3 | 50.9 | 42.2 |
| FT on LAION-4M-prep + neg. cap. | 32.0 | 0.0 | 0.0 | 26.5 | 0.0 | 0.0 | 39.9 | 48.9 | 47.3 | 45.7 | 40.1 |
| *RACA-CLIP (Ours)* | 34.46 | 4.85 | 0.0 | 37.5 | 5.39 | 0.0 | 50.24 | 55.23 | 55.66 | 5119 | 47.4 |
| Random chance | 25.0 | 6.3 | 0.4 | 25.0 | 6.3 | 0.4 | 50.0 | 50.0 | 50.0 | 50.0 | 41.7 |

annotate relations in captions, which limits compositional coverage. Our triplet-based objective is well-suited for such cases, but its effectiveness depends on having diverse relational supervision; with richer relation diversity, relation-aware modeling can substantially improve performance on benchmarks like What'sUp.

### E.5 SCORES ON WINOGROUND BENCHMARK

Table 15: **Winoground:** Accuracy (%) on text, image, group, and their mean (Avg.). For each column, the best result is in **bold** and the second best is underlined.

| Method | Text | Image | Group | Avg. |
|---|---|---|---|---|
| CLIP | 30.7 | 10.5 | 8.0 | 16.4 |
| NegCLIP | 30.5 | 10.5 | 8.0 | 16.3 |
| TripletCLIP | 26.8 | **11.2** | 7.0 | 15.0 |
| CE-CLIP | 18.5 | **11.2** | 5.0 | 11.6 |
| *RACA-CLIP (Ours)* | **34.8** | 11.0 | 9.2 | **18.3** |

Winoground evaluates challenging compositional disambiguation through paired image–caption setups. Using the same CLIP-like backbone and evaluation protocol, we report accuracy on the text, image, and group subtasks, along with the overall average (Table 15). RACA-CLIP attains the highest average score (18.3%) among the CLIP variants considered, with improvements on both the text and group metrics. Notably, this gain is achieved without any form of hard-negative mining, indicating that the region- and triplet-aware supervision learned during fine-tuning also transfers to the fine-grained disambiguation required by Winoground.

### E.6 SCORES ON SUGARCREPE++ BENCHMARK

Table 16: **SugarCrepe++:** Compositional accuracy (%) on replace and swap perturbations. For each column, the best result is in **bold** and the second best is underlined.

| Method | Replace | | | Swap | | Avg. |
|---|---|---|---|---|---|---|
| | Obj | Attr | Rel | Obj | Attr | |
| CLIP | 87.89 | 68.02 | 50.14 | 37.95 | 48.19 | 58.4 |
| NegCLIP | 89.58 | 69.54 | 52.43 | **54.69** | **58.40** | 64.9 |
| TripletCLIP | 86.98 | 71.82 | **62.23** | 39.59 | 48.49 | 61.8 |
| CE-CLIP | 81.84 | 62.43 | 53.12 | 40.00 | 40.39 | 55.6 |
| *RACA-CLIP (Ours)* | **91.88** | **74.49** | 59.53 | 44.48 | 55.10 | **65.1** |

SugarCrepe++ extends SugarCrepe by pairing each image with two semantically correct but lexically diverse captions, along with a strong hard negative caption. The benchmark evaluates whether both positives consistently rank above the negative under several compositional perturbations (Table 16). RACA-CLIP achieves the highest average accuracy (65.1%), outperforming others across replace and swap categories. Notably, SugarCrepe++ is designed to mirror natural human descriptions, where multiple paraphrastic interpretations of the same scene are equally valid. RACA-CLIP's

strong performance therefore suggests that its structured supervision enables more human-aligned cross-modal reasoning, maintaining stable alignment across diverse yet semantically consistent captions.

## E.7 Detailed Scores on SugarCrepe benchmark

Table 17: **SugarCrepe results.** Accuracy (*Replace* (Obj/Att/Rel), *Swap* (Obj/Att), *Add* (Obj/Att). *Ours* achieves consistent improvements over CLIP across all categories. Bold indicates the best score and underline indicates the second-best.

| Methods | Replace | | | Swap | | Add | |
|---|---|---|---|---|---|---|---|
| | Object | Attribute | Relation | Object | Attribute | Object | Attribute |
| Human | 100 | 99 | 97 | 99 | 100 | 99 | 99 |
| **Pretrained models** | | | | | | | |
| *Text-only model* | | | | | | | |
| Vera | 49.39 | 49.62 | 49.36 | 49.19 | 49.40 | 49.42 | 49.57 |
| Grammar | 50.00 | 50.00 | 50.00 | 50.00 | 50.00 | 50.00 | 50.00 |
| *OpenAI* | | | | | | | |
| ViT-B-32 | 90.92 | 80.08 | 69.20 | 61.38 | 63.96 | 77.21 | 68.79 |
| ViT-L-14 | 94.07 | 79.19 | 65.15 | 60.16 | 62.31 | 78.32 | 71.53 |
| *DataComp* | | | | | | | |
| ViT-B-32 (13M) | 56.90 | 56.85 | 51.99 | 50.81 | 50.00 | 53.93 | 60.55 |
| ViT-B-32 (128M) | 77.00 | 69.54 | 57.68 | 57.72 | 57.06 | 66.73 | 64.88 |
| ViT-B-16 (1B) | 92.68 | 79.82 | 63.94 | 56.10 | 57.66 | 84.34 | 78.61 |
| ViT-L-14 (13B) | 95.52 | 84.52 | 69.99 | 65.04 | 66.82 | 91.03 | 84.97 |
| *Other datasets* | | | | | | | |
| LaCLIP + HN (CC-3M) | 63.44 | 55.96 | 50.71 | 50.60 | 48.57 | 56.98 | 51.16 |
| LaCLIP (CC-12M) | 75.06 | 65.48 | 58.68 | 53.47 | 57.66 | 67.65 | 66.76 |
| CLIP (CC-12M) | 85.77 | 79.18 | 64.51 | 61.78 | 58.71 | 74.24 | 68.35 |
| FLIP (CC-12M) | 84.07 | 75.88 | 66.00 | 60.16 | 61.56 | 71.67 | 63.15 |
| FLIP$_{Attn-0.3}$(CC-12M) | 86.07 | 75.00 | 62.23 | 60.57 | 59.16 | 74.68 | 68.64 |
| IL-CLIP (CC-3M) | | Avg. 67.0 | | | Avg. 66.0 | | Avg. 66.0 |
| IL-CLIP (CC-12M) | | Avg. 73.0 | | | Avg. 62.9 | | Avg. 73.8 |
| Codebook-CLIP (CC-3M) | | Avg. 64.8 | | | Avg. 54.9 | | Avg. 65.9 |
| Codebook-CLIP (CC-12M) | | Avg. 71.1 | | | Avg. 59.5 | | Avg. 71.3 |
| NegCLIP (CC-12M) | | Avg. 70.2 | | | Avg. 67.2 | | Avg. 65.0 |
| SLIP (YFCC-15M) | | Avg. 75.2 | | | Avg. 58.6 | | Avg. 73.7 |
| SF-CLIP (YFCC-12M) | | Avg. 77.3 | | | Avg. 61.6 | | Avg. 74.8 |
| SPARO (LAION-400M) | | Avg. 80.9 | | | Avg. 65.2 | | Avg. 83.0 |
| CLIP | 90.92 | 80 | 69.13 | 61.2 | 64 | 77.16 | 68.2 |
| **Finetuned models** | | | | | | | |
| CLIP-FT | 90.92 | 79.69 | 64.01 | 60.82 | 64.26 | 84.67 | 78.76 |
| NegCLIP | 91.53 | 83.25 | 73.97 | **72.24** | 67.72 | 86.95 | 88.44 |
| CoN-CLIP | 93.58 | 80.96 | 63.30 | 59.18 | 65.16 | 87.29 | 79.62 |
| BLIP-SVGL | 53.69 | 52.41 | 47.43 | 44.89 | 56.00 | 45.87 | 50.57 |
| TSVLC (RB) | 91.34 | 81.34 | 64.15 | 68.16 | 69.07 | 79.49 | 91.33 |
| TSVLC (LLM+RB) | 88.13 | 76.78 | 62.73 | 64.08 | 66.67 | 75.80 | 81.07 |
| LaCLIP | 93.28 | 81.09 | 61.73 | 62.44 | 58.70 | 81.57 | 73.55 |
| CyCLIP | 80.87 | 66.12 | 57.54 | 53.88 | 52.11 | 71.48 | 65.75 |
| TripletCLIP | 94.43 | **85.53** | **80.94** | 69.80 | 69.82 | 90.40 | 86.27 |
| Cluster-Mask-CLIP | 86.13 | 75.13 | 64.65 | 66.67 | 63.36 | 74.92 | 71.24 |
| *RACA-CLIP (Ours)* | **94.73** | 85.15 | 72.11 | 71.00 | **70.50** | **93.40** | **93.06** |
| w.r.t CLIP | (+3.81) | (+5.15) | (+2.98) | (+9.80) | (+6.50) | (+16.24) | (+24.86) |

## E.8    CAUSAL AND PROBING ANALYSIS

Table 18: **Effect of region- and triplet-level losses.** We report average accuracies (%) over compositional benchmarks (WhatsUp, COLA, Winoground, SugarCrepe, SugarCrepe++) and the fine-grained MMVP-VLM diagnostic, under different loss configurations. ✓ indicates that the corresponding loss is used. For each column, the best result is in **bold** and the second best is underlined.

| Model | Region | Triplet | WhatsUp | COLA | Wino-ground | Sugar-Crepe | SugarCrepe++ | MMVP-VLM | Avg. |
|---|---|---|---|---|---|---|---|---|---|
| CLIP | | | 41.8 | 22.8 | 16.4 | 72.9 | 54.6 | 27.39 | 39.32 |
| CLIP-GBC (global) | | | 41.9 | 27.9 | 17.1 | 77.6 | 57.6 | 28.89 | 41.83 |
| CLIP-GBC + Reg. | ✓ | | 44.6 | 30.6 | 17.8 | 80.7 | 63.27 | 28.89 | 44.31 |
| CLIP-GBC + Tri. | | ✓ | 43.4 | **33.0** | 15.41 | 81.0 | 62.20 | 30.37 | 44.23 |
| *RACA-CLIP (Ours)* | ✓ | ✓ | **47.0** | 30.63 | **18.3** | **83.0** | **65.1** | **33.3** | **46.22** |

To disentangle the contribution of structure-aware learning from the effect of training on GBC data alone, we perform an ablation in which we first fine-tune a CLIP model using only global image–caption pairs from GBC (**CLIP-GBC (global)**), and then incrementally add the region-level and triplet-level losses. Table 18 reports results across all compositional benchmarks, including the MMVP-VLM diagnostic.

Both structured objectives individually improve over the CLIP-GBC baseline: adding only the region-level loss increases average accuracy from 41.83% to 44.31%, and adding only the triplet-level loss yields a similar improvement to 44.23%. When combined, these two objectives produce the full RACA-CLIP model, which achieves the highest overall accuracy (46.22%). The gains consistently appear on benchmarks that require attribute binding, relation reasoning, and multi-positive alignment, indicating that the improvements stem directly from enforcing region- and relation-aware alignment rather than from data effects alone. This provides causal evidence that the structured losses are responsible for the additional compositional capacity observed in RACA-CLIP.

## E.9    EFFECT OF BACKBONE

Table 19: **Effect of backbone capacity.** We report average accuracies (%) over compositional benchmarks (WhatsUp, COLA, Winoground, SugarCrepe, SugarCrepe++) and the fine-grained MMVP-VLM diagnostic, for CLIP and RACA-CLIP with ViT-B/32 and ViT-L/14 backbones. For each backbone block, the best method is in **bold**.

| Backbone | Method | WhatsUp | COLA | Wino-ground | Sugar-Crepe | SugarCrepe++ | MMVP-VLM | Avg. |
|---|---|---|---|---|---|---|---|---|
| ViT-B/32 | OPEN CLIP | 41.8 | 22.8 | 16.4 | 72.9 | 54.6 | 27.4 | 39.3 |
| ViT-B/32 | *RACA-CLIP (Ours)* | **47.0** | **30.6** | **18.3** | **83.0** | **65.1** | **33.3** | **46.2** |
| ViT-L/14 | OPEN CLIP | 42.9 | 27.0 | 15.5 | 80.16 | 60.88 | 31.9 | 43.1 |
| ViT-L/14 | *RACA-CLIP (Ours)* | **48.9** | **32.86** | **19.5** | **86.17** | **66.18** | **36.3** | **48.3** |

To assess whether our structured supervision continues to provide benefits as model capacity increases, we repeat our experiments with a **ViT-L/14** backbone using the same LoRA-based fine-tuning recipe and only light hyperparameter sanity checks (e.g., learning-rate scaling), without extensive tuning. Table 19 reports average accuracies across compositional benchmarks and MMVP-VLM for both CLIP and RACA-CLIP with **ViT-B/32** and **ViT-L/14** backbones.

For ViT-B/32, RACA-CLIP improves the OpenCLIP baseline from 39.3% to 46.2% average accuracy. For ViT-L/14, we again observe consistent gains, from 43.1% to 48.3%. These improvements are obtained under essentially the same training setup and without aggressive hyperparameter tuning for the larger backbone, suggesting that they are a conservative estimate of the achievable performance. Overall, the results indicate that the benefits of region- and triplet-aware supervision persist and even amplify with increased backbone capacity, rather than being a small-model artifact.

## E.10 WHY RACA-CLIP AVOIDS HARD-NEGATIVE–DRIVEN TRAINING

A central motivation for our formulation is the inherent *information imbalance* (Schrodi et al. (2025)) between the two modalities in standard CLIP-style training. Captions explicitly encode objects, attributes, and relations, whereas images provide these structures only implicitly through spatial layout and appearance. A seemingly natural way to "resolve" this imbalance is to engineer increasingly descriptive captions and to generate families of hard negatives around them (e.g., via templates or large language models), so that the model learns to separate fine-grained textual variants. However, such strategies operate almost entirely on the *text* side of the joint space. Methods that rely heavily on hard textual negatives attempt to correct this imbalance by aggressively separating near-miss captions (e.g., swapped objects, attributes, or roles), but prior analyses show that this leads to an *asymmetric embedding geometry*: the text manifold becomes well-separated, yet the image manifold remains under-constrained, leaving visually fine-grained concepts entangled and producing unstable image-text retrieval behavior (Miranda et al. (2024)).

RACA-CLIP addresses this imbalance structurally. Rather than pushing images away from synthetically corrupted captions, we impose visually grounded constraints through *region-aligned multi-positives* and *relation-aware triplets*. These losses explicitly "pull" images toward their own part-level structure—subregions that satisfy IoU alignment and triplets that enforce role-specific consistency. This creates *bidirectional geometric constraints* that tighten the image manifold, disentangle object- and attribute-level variations, and yield a more stable and symmetric joint metric space. In effect, RACA-CLIP strengthens the visual side of the representation, which hard-negative–based formulations typically leave weakly supervised.

This geometric improvement is corroborated by our analysis on **UT-Zappos-50K** (Section 4.6), a dataset known for subtle attribute distinctions (e.g., shape vs. style, material vs. texture). On this benchmark, RACA-CLIP produces clearer separation between closely related attribute pairs, while hard-negative–driven variants exhibit significantly weaker gains. This aligns with our central claim: improvements derived from linguistic perturbations do not reliably translate to visually grounded distinctions. By grounding supervision directly in visual evidence rather than synthetic caption edits, RACA-CLIP yields a more robust, balanced, and compositional embedding space across both modalities.

## E.11 MORE QUALITATIVE EXAMPLES

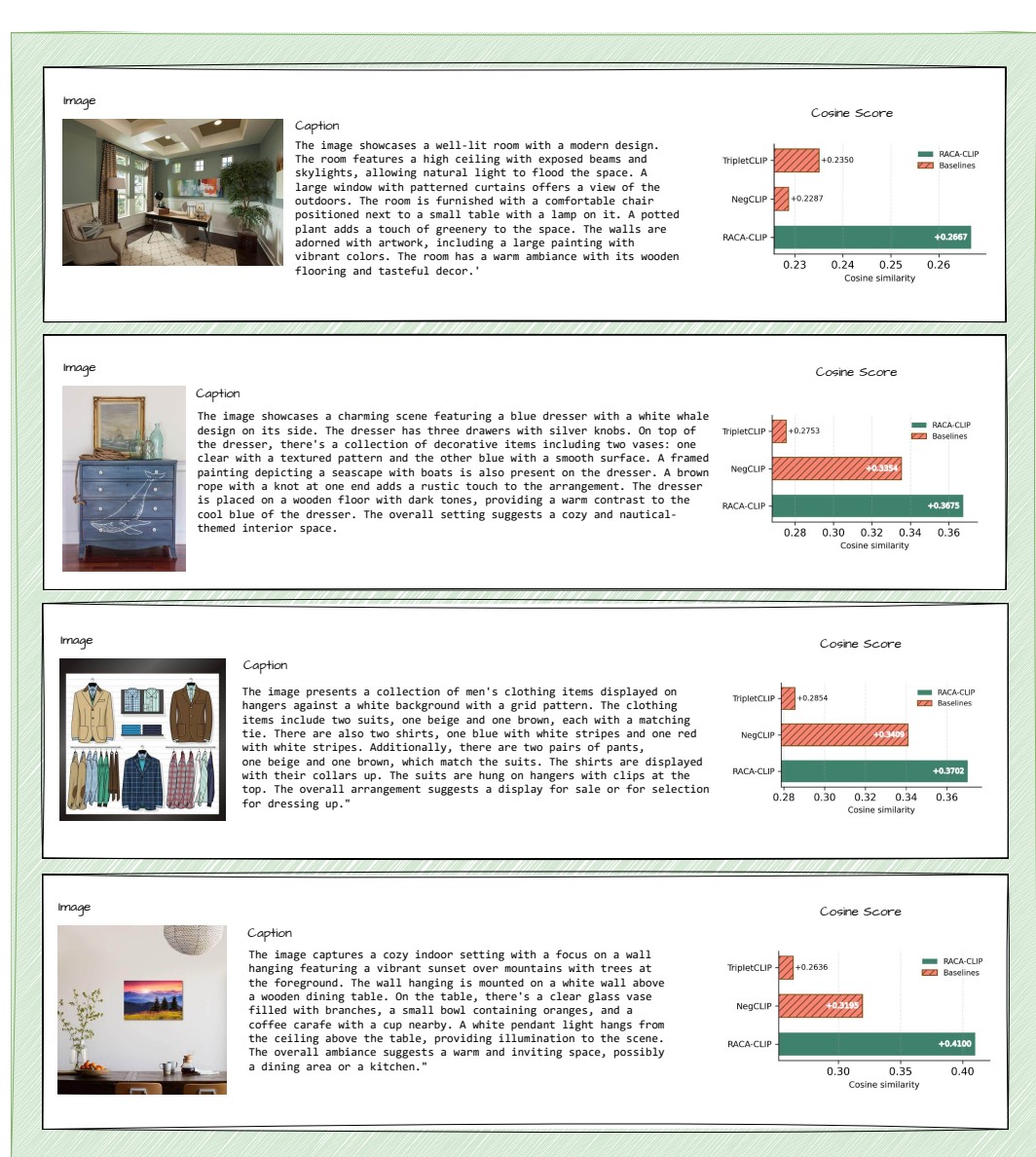

Figure 4: **Dense caption alignment across contrastive objectives.** Each row shows an image, its dense caption, and the cosine similarity between the image and that caption for **RACA-CLIP**, **Neg-CLIP**, and **TripletCLIP**. The examples cover diverse indoor scenes with multiple objects, attributes, and relations, where dense-caption alignment is challenging. RACA-CLIP consistently assigns the highest similarity to the dense caption, indicating improved grounding of fine-grained compositional details beyond what hard-negative–based objectives (NegCLIP, TripletCLIP) achieve.

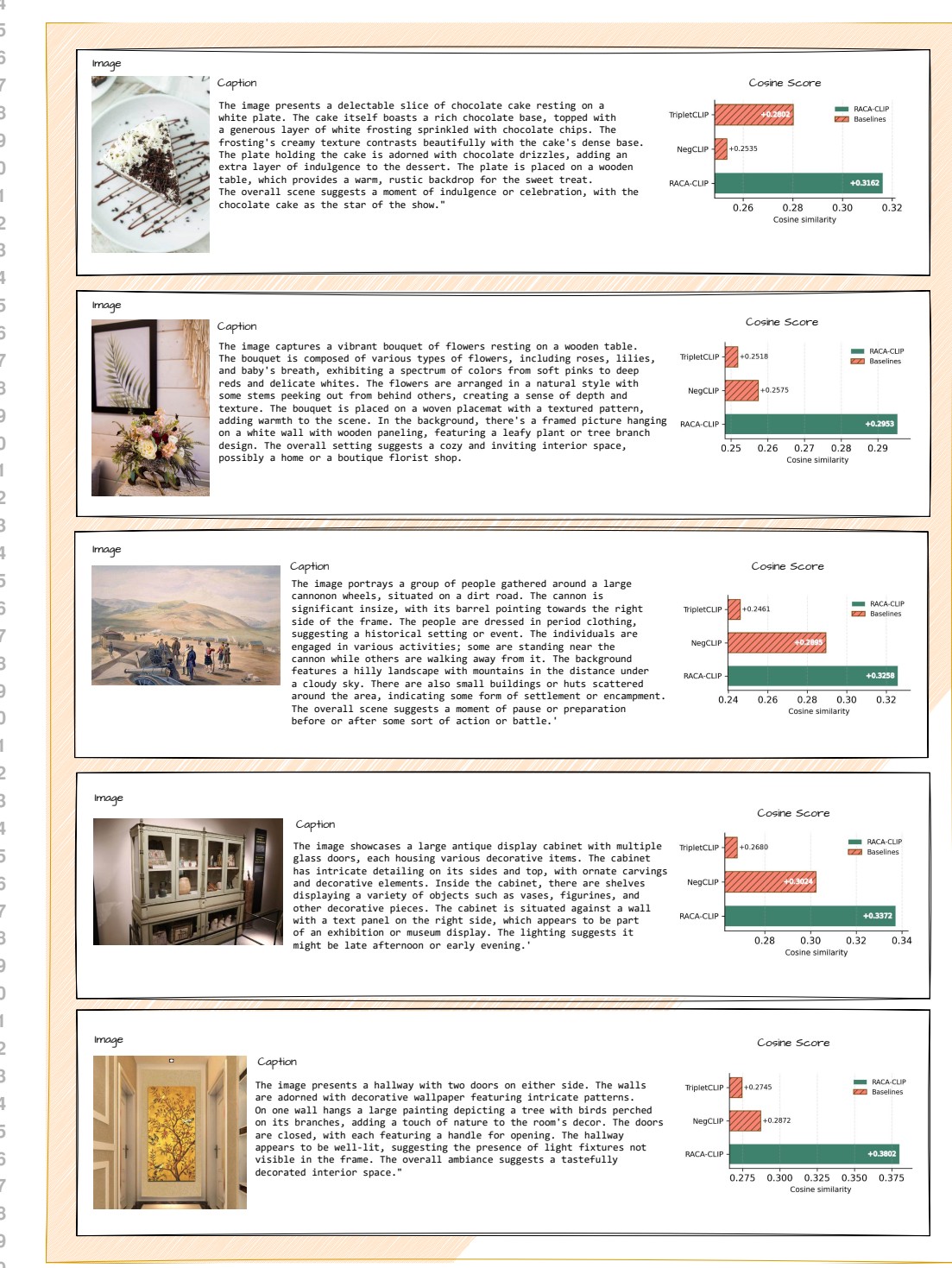

Figure 5: **Dense caption alignment across contrastive objectives.** Each row shows an image, its dense caption, and the cosine similarity between the image and that caption for **RACA-CLIP**, **NegCLIP**, and **TripletCLIP**. The examples span indoor, outdoor, and cluttered scenes with multiple entities and relations. RACA-CLIP systematically yields the highest similarity for these dense captions, suggesting that region- and relation-aware supervision improves alignment with complex compositional descriptions beyond what hard-negative–based objectives (NegCLIP, TripletCLIP) achieve.