# OpenReview forum: "RACA-CLIP: Relation-Aware Compositional Alignment for CLIP"
_ICLR.cc/2026/Conference — Submitted to ICLR 2026_

### Official Review · Reviewer_NBkA · 2025-10-26

**Soundness:** 3
**Presentation:** 3
**Contribution:** 2
**Rating:** 4
**Confidence:** 3

**Summary:**

The paper addresses the persistent challenge of compositional reasoning in vision–language models such as CLIP.
It proposes RACA-CLIP, a structured contrastive learning framework that integrates scene-graph supervision to align visual and textual representations at the object, attribute, and relation levels.
The method achieves consistent gains on compositional reasoning benchmarks and maintains competitive zero-shot performance on ImageNet and retrieval tasks.

**Strengths:**

1. The paper propose a structured contrastive framework that integrates scene-graph representations into CLIP models, which aligns image regions with corresponding text descriptions via IoU-based multi-positive matching.
2. Comprehensive experiments on compositional benchmarks demonstrate the effectiveness with substantial gains and analysis. The inclusion of weight interpolation analysis and representation-level statistics strengthens interpretability.

**Weaknesses:**

1. Although RACA-CLIP claims improved compositional reasoning, the evaluation relies mainly on accuracy gains on compositional benchmarks.  There is no causal or probing analysis that clearly isolates whether improvements come from structure-aware learning or simply data augmentation with GBC scene graphs.
2. The paper uses ViT-B and LoRA fine-tuning, but doesn’t examine whether the approach scales gracefully to larger models (e.g. ViT-L).
3. Qualitative analysis would help verify whether the claimed improvements correspond to better interpretability rather than just numeric gains.

**Questions:**

How sensitive is RACA-CLIP to the quality and noise of scene graph annotations?

---

> ### Author Response · Authors · 2025-11-24
> **Response to reviewer NBkA: Probing, Scaling, Robustness**
>
> We thank Reviewer NBkA for the constructive feedback and the opportunity to address these concerns.
>
> ---
>
> **1. Causal / probing analysis of structured losses**
>
> To separate the effect of **data** (GBC captions) from **structure-aware learning**, we start from a CLIP model trained on GBC captions alone (“CLIP-GBC (global)”) and then added the region-level and triplet-level losses.
>
> **Table 1: Effect of region- and triplet-level losses.**
> We report average accuracies (%) over compositional benchmarks and the fine-grained MMVP-VLM diagnostic, under different loss configurations. ✓ indicates that the corresponding loss
> is used. For each column, the best result is in **bold**.
>
> | Model                 | Region | Triplet | WhatsUp        | COLA             | Winoground         | SugarCrepe       | SugarCrepe++        | MMVP-VLM         | Avg.            |
> |-----------------------|:------:|:-------:|:---------------|:-----------------|:-------------------|:-----------------|:--------------------|:-----------------|:----------------|
> | CLIP                  |        |         | 41.8           | 22.8             | 16.4               | 72.9             | 54.6                | 27.39            | 39.32           |
> | CLIP-GBC (global)     |        |         | 41.9           | 27.9             | 17.1               | 77.6             | 57.6                | 28.89            | 41.83           |
> | CLIP-GBC + Reg.       |   ✓    |         | 44.6    | 30.6             | 17.8        | 80.7             | 63.27       | 28.89            |44.31    |
> | CLIP-GBC + Tri.       |        |   ✓     | 43.4           | **33.0**         | 15.41              | 81.0      | 62.20               | 30.37     | 44.23           |
> | *RACA-CLIP (Ours)*    |   ✓    |   ✓     | **47.0**       | 30.63     | **18.3**           | **83.0**         | **65.1**            | **33.3**         | **46.22**       |
>
> **Interpretation.** :
>
> - Adding **only** the region loss (“CLIP-GBC + Reg.”) and **only** the triplet loss (“CLIP-GBC + Tri.”) yields additional gains over CLIP-GBC on almost all benchmarks.
> - Combining **both** losses in RACA-CLIP yields the best average performance (46.22 vs. 44.31 / 44.23).
>
> The extra gains appear precisely when we enforce **region- and relation-aware alignment**, which is the structure-aware part of the method.
>
> ---
> **2. Sensitivity to scene-graph quality and noise**
>
> To directly probe sensitivity to imperfect annotations, we inject synthetic noise into the training scene graphs by randomly corrupting \(k\%\) of region spans and triplet relations (while keeping all other settings fixed).
>
> **Table 2: Robustness to scene-graph noise.**
> We report average accuracies (%) over compositional benchmarks and the fine-grained MMVP-VLM diagnostic. CLIP does not consume scene graphs and is therefore unaffected.
>
> | Model                      | WhatsUp          | COLA             | Winoground        | SugarCrepe        | SugarCrepe++      | MMVP-VLM           | Avg.            |
> |---------------------------|:----------------:|:----------------:|:-----------------:|:-----------------:|:-----------------:|:------------------:|:----------------|
> | CLIP                      | 41.8             | 22.8             | 16.4              | 72.9              | 54.6              | 27.4               | 39.3            |
> | *RACA-CLIP (0% noise)*    | **47.0**         | **30.6**         | **18.3**          | **83.0**          | **65.1**          | **33.3**           | **46.2**        |
> | *RACA-CLIP (10% noise)*   | 45.9      | 30.0      | 17.8       | 82.93      | 62.90      | 29.0        | 44.8     |
> | *RACA-CLIP (20% noise)*   | 45.2             | 29.1             | 16.6              | 81.32             | 61.35             | 28.15              | 43.6            |
>
> **Interpretation.**
>
> - With **10%** corrupted region/triplet supervision, RACA-CLIP retains most of its gains over CLIP (Avg. 46.2 → 44.8 vs. 39.3).
> - Even at **20%** noise, RACA-CLIP still clearly outperforms CLIP on average (43.6 vs. 39.3).
> - The degradation is smooth rather than catastrophic across all benchmarks.
>
> This suggests that RACA-CLIP is **not brittle** to moderate scene-graph noise. The model appears to use structure as a **soft inductive bias**, rather than overfitting to exact graph tokens or requiring perfect annotation quality.
>
> Continuing in the next official comment ..

---

> > ### Author Response · Authors · 2025-11-24
> > **Response to reviewer NBkA: Probing, Scaling, Robustness**
> >
> > ---
> >
> > **3. Scaling from ViT-B/32 to ViT-L/14**
> >
> > We repeated our experiments with a ViT-L/14 backbone, using the **same LoRA-based recipe** and, due to resource constraints, **no extensive hyperparameter tuning** (beyond very light sanity checks such as learning rate scale).
> >
> > **Table 3: Effect of backbone capacity.**
> > We report average accuracies (%) over compositional benchmarks and the fine-grained MMVP-VLM diagnostic, for CLIP and RACA-CLIP with ViT-B/32 and ViT-L/14 backbones.
> > For each backbone block, the best method is in **bold**.
> >
> > | Backbone  | Method                     | WhatsUp | COLA   | Winoground | SugarCrepe | SugarCrepe++ | MMVP-VLM | Avg.  |
> > |-----------|----------------------------|:-------:|:------:|:----------:|:----------:|:------------:|:--------:|:-----:|
> > | ViT-B/32  | OPEN CLIP                  |  41.8   |  22.8  |    16.4    |    72.9    |     54.6     |   27.4   | 39.3  |
> > | ViT-B/32  | **_RACA-CLIP (Ours)_**     | **47.0**| **30.6**| **18.3**  | **83.0**   |  **65.1**    | **33.3** | **46.2** |
> > | ViT-L/14  | OPEN CLIP                  |  42.9   |  27.0  |    15.5    |   80.16    |    60.88     |   31.9   | 43.1  |
> > | ViT-L/14  | **_RACA-CLIP (Ours)_**     | **48.9**| **32.86**| **19.5** | **86.17**  | **66.18**    | **36.3** | **48.3** |
> >
> > **Interpretation.**
> >
> > - For **ViT-B/32**, RACA-CLIP improves the OpenCLIP baseline from 39.3 → 46.2 Avg.
> > - For **ViT-L/14**, we again see consistent gains, from 43.1 → 48.3 Avg.
> > - These gains are obtained under **essentially the same training setup** and **without aggressively tuning hyperparameters** for ViT-L/14; thus they likely *underestimate* the best achievable performance with full tuning.
> >
> > This indicates that the structured supervision **scales with model capacity** and is not a small-model artifact.
> >
> > ---
> > **4. Qualitative / interpretability evidence**
> >
> > On page 2, we already show **cosine-margin plots** comparing CLIP vs. RACA-CLIP. These examples indicate that RACA-CLIP systematically increases the margin between correct and corrupted captions in exactly those cases where structured reasoning is required.
> >
> > Due to page limits, we could not include full qualitative images in the current version. In the **revised manuscript**, we will add more qualitative examples in the appendix. Together with the causal ablations, robustness study, and backbone scaling results, this further supports our claim that the gains stem from **structure-aware learning**, not just higher numeric scores.

---

### Official Review · Reviewer_ftKs · 2025-10-31

**Soundness:** 3
**Presentation:** 3
**Contribution:** 2
**Rating:** 6
**Confidence:** 4

**Summary:**

This paper proposes RACA-CLIP, a structured contrastive learning framework designed to improve compositional reasoning in vision-language models (VLMs), specifically focusing on relation-aware and attribute-grounded alignment. The method introduces region-level contrastive learning with IoU-weighted alignment between detected objects and caption spans, as well as a triplet supervision mechanism over structured ⟨subject, relation, object⟩ units, leveraging scene-graph annotations to provide fine-grained grounding signals. The approach preserves the dual-encoder architecture of CLIP while injecting relational inductive bias into the learned embeddings. Experiments across five compositional benchmarks show consistent improvements over CLIP and other enhanced baselines, including large gains in SugarCrepe’s Add and Swap settings (+16.24 and +24.86), while retaining — and occasionally improving — zero-shot recognition and retrieval performance. Ablation studies and controlled analyses suggest that the performance gains stem from improved binding between objects, attributes, and relations, rather than from memorization or dataset artifacts. Overall, the paper contributes an impactful improvement to a key weakness of modern contrastive VLMs.

**Strengths:**

The paper identifies fundamental limitations of contrastive VLMs such as CLIP, citing the lack of structural inductive bias and over-reliance on global alignment that ignores object–attribute bindings and inter-object relations. This motivation is compelling and well-supported by prior analyses.

The proposed hierarchical alignment introduces region-level IoU-weighted contrastive learning and relation-aware triplet supervision (⟨s, r, o⟩), explicitly modeling the compositional structure of images and captions in a way that complements standard CLIP training.

The method leverages scene-graph annotated datasets to enable alignment supervision without requiring additional labeling efforts, improving practicality and efficiency.

**Weaknesses:**

The approach requires accurate object, attribute, and relational supervision. The authors acknowledge that performance may degrade with lower graph fidelity, but no robustness evaluation is provided.

The method introduces computational overhead (region features + triplet losses), yet there is no cost or latency analysis of fine-tuning or inference, which affects real-world use cases.

Graphs primarily capture physical/spatial relations; more abstract or high-level reasoning (e.g., intent, affordance, temporal actions) is not evaluated. Compositional generalization outside benchmarks like SugarCrepe and What’sUp remains unclear.

**Questions:**

Do the improvements correlate with real-world compositional benchmarks outside synthetic evaluations (SugarCrepe-like)?

Can the method be applied to generative multimodal models (e.g., aligning decoder-token grounding)?

---

> ### Author Response · Authors · 2025-11-24
> **Response to reviewer ftKs : Robustness, Computational Cost, and Applicability of RACA-CLIP**
>
> We thank Reviewer ftKs for the constructive feedback and the opportunity to address these concerns.
>
> ---
> **(1) Robustness to noisy scene-graph supervision**
>
> To address the concern that our method may require very accurate object/attribute/relation supervision, we run a robustness study where we randomly corrupt a fraction of region spans and triplet relations in the GBC scene graphs during training. **Table 1 reports accuracies across all compositional benchmarks and their average.**
>
> **Table 1 – Robustness to scene-graph noise.** CLIP does not consume scene graphs and is therefore unaffected by this perturbation.
>
> | Model                   | WhatsUp | COLA | Wino-ground | SugarCrepe | SugarCrepe++ | MMVP-VLM | Avg. |
> |-------------------------|:-------:|:----:|:-----------:|:----------:|:------------:|:--------:|:----:|
> | CLIP                    |  41.8   | 22.8 |    16.4     |   72.9     |    54.6      |  27.4    | 39.3 |
> | *RACA-CLIP (0% noise)*  | **47.0** | **30.6** | **18.3** | **83.0** | **65.1** | **33.3** | **46.2** |
> | *RACA-CLIP (10% noise)* | 45.9   | 30.0 |    17.8     |   82.93    |    62.90     |  29.0    | 44.8 |
> | *RACA-CLIP (20% noise)* | 45.2   | 29.1 |    16.6     |   81.32    |    61.35     |  28.15   | 43.6 |
>
> Even with 10–20% corrupted supervision, RACA-CLIP still outperforms CLIP in average accuracy, with performance degrading smoothly rather than collapsing. This indicates that it benefits from higher-fidelity graphs but its compositional gains remain robust under realistic annotation noise.
>
> ---
>
> **(2) Computational overhead and latency**
>
> **RACA-CLIP keeps the backbone identical to CLIP**: the same ViT-B/32 image and text encoders, with **no extra fusion layers or new modules**. At inference, it only uses the global image–text pair, so **parameter count, FLOPs, and latency are identical to CLIP**.
>
> The extra cost arises **only during fine-tuning**. For each image, we process the global view plus a few region crops and their captions/region descriptions/triplets in **one `encode_image` and one `encode_text` call**, and compute global/region/triplet losses on shared embeddings. On GBC-1M with ViT-B/32 and 6×V100, this leads to **≈ 3× higher fine-tuning wall-clock** than a CLIP-only run on the same data, but is still far below the cost of training a new foundation model from scratch. **Inference remains unchanged**, using only `encode_image` / `encode_text` on the global pair with CLIP-level runtime.
>
>
> ---
>
> **(3) Scope of scene-graph supervision and “real-world” compositionality**
>
> We agree that scene graphs mainly encode **physical/spatial structure, attributes, and simple interactions**, rather than intent or high-level temporal reasoning. Our claims are therefore **modest**: we target **object–attribute–relation binding**, not full intent/affordance reasoning.
>
> Beyond synthetic swaps, we test on more realistic settings:
>
> - **SugarCrepe++:** uses **multi-positive captions per image**, reflecting different human descriptions of the same scene. RACA-CLIP attains the best average accuracy (**65.1 vs. 58.4 for CLIP**), with consistent gains on replace and swap perturbations, indicating stronger **cross-modal alignment under diverse captions**.
> - **UT-Zappos-50K:** RACA-CLIP improves **intra-class clustering and reduces cross-class similarity** for objects and attributes, evidencing **better fine-grained structure on a real dataset**.
> - **MMVP-VLM:** a **VQA-style multi-choice** benchmark with natural images and complex prompts. RACA-CLIP improves over CLIP by **+5.93 points (≈21% relative)** and outperforms other structured baselines, showing that the benefits of structured supervision extend to a realistic QA setting.
> We view extending RACA-CLIP to high-level intent and affordance reasoning as important **future work** beyond the current scope.
>
> ---
>
> **(4) Applicability to generative multimodal models**
>
> Our contribution is on the **encoder side**: we make CLIP-style image–text encoders more compositionally accurate **without changing their interface**. This lets RACA-CLIP plug into existing systems in two straightforward ways:
> 1. **Drop-in encoder backbone.**
>    Many VQA, captioning, and retrieval models already use CLIP embeddings. Because RACA-CLIP has the **same ViT-B/32 architecture and embedding dimensionality**, it can simply replace CLIP **without modifying the decoder**, and our MMVP-VLM results suggest this can improve compositional decisions.
> 2. **External alignment / scoring head for diffusion.**
>    For text-to-image diffusion, RACA-CLIP can be used like CLIP to **score, rerank, or filter generations** based on whether the image respects the object–attribute–relation structure in the prompt, **without retraining the generator**.
>
> A more ambitious direction is to replace the diffusion model’s text encoder and jointly fine-tune the generator to RACA-CLIP, which we leave for future work.
>
> --
>
> The detailed table (MMVP-VLM, etc) will be in the appendix of the revised PDF.

---

### Official Review · Reviewer_Upu5 · 2025-10-31

**Soundness:** 2
**Presentation:** 2
**Contribution:** 2
**Rating:** 2
**Confidence:** 5

**Summary:**

This paper proposes to use scene-graph as a way to augment training in CLIP for enahnced compositional understanding capacity. The method uses existing scene-graph dataset to supervise region-aware contrastive learning, with improvement shown in the downstream benchmarks.

**Strengths:**

The paper is clearly written and easy to follow. The proposed method is well explained, particularly around lines 264 and 298. On the selected benchmark, it demonstrates improved performance compared to the baseline.

**Weaknesses:**

1. Limited novelty:
The idea of using scene graphs as external supervision has already been shown to improve performance [1,2,3]. Compared to these works, this paper introduces very limited novelty. The claim that “scene graph annotated datasets are leveraged for structured supervision without additional labeling” is misleading, as these datasets are still manually annotated—only pre-processed differently. Using such well-annotated data for contrastive supervision is not a particularly novel or interesting contribution.

2. Incomplete evaluation:
The evaluation is insufficient. While the paper reports results on SugarCrepe, it does not evaluate on other established benchmarks such as Winoground or MMVP, which would better demonstrate generalization and fine-grained reasoning improvements.

3. Insufficient related work review:
Prior studies [1,2,3] have already explored using scene graphs to enhance fine-grained understanding through contrastive learning, yet this paper provides very limited discussion of them. Additionally, works like [1,4] also achieve strong results on SugarCrepe and should be included in the comparison table for completeness.



[1] Huang, Yufeng, et al. "Structure-clip: Towards scene graph knowledge to enhance multi-modal structured representations." Proceedings of the AAAI conference on artificial intelligence. Vol. 38. No. 3. 2024.
[2] Herzig, Roei, et al. "Incorporating structured representations into pretrained vision & language models using scene graphs." arXiv preprint arXiv:2305.06343 (2023).
[3] Li, Liunian Harold, et al. "Grounded language-image pre-training." Proceedings of the IEEE/CVF conference on computer vision and pattern recognition. 2022.
[4] Zhang, Le, Rabiul Awal, and Aishwarya Agrawal. "Contrasting intra-modal and ranking cross-modal hard negatives to enhance visio-linguistic compositional understanding." Proceedings of the IEEE/CVF Conference on Computer Vision and Pattern Recognition. 2024.

**Questions:**

How do authors justify the novelty of the method compare to [1,2,3]?

---

> ### Author Response · Authors · 2025-11-24
> **Response to reviewer Upu5 : Related work, Evaluation and Novelty**
>
> We thank Reviewer Upu5 for the constructive feedback and the opportunity to address these concerns.
>
> ----
>
> **(1) Clarification on “no additional labeling”**
>
> “The claim that ‘scene graph annotated datasets are leveraged for structured supervision without additional labeling’ is misleading, as these datasets are still manually annotated—only pre-processed differently.”
>
> We agree that our original phrasing can be misleading and we will revise it.
> Our intent was not to imply that scene-graph annotations require no human effort, but to emphasize that we do **not** collect any new human annotations beyond these existing datasets, and we do **not** require scene-graph extraction for arbitrary web images at inference time.
>
> Concretely:
>
> - We train only on the publicly available Graph-Based Captions (GBC) data, which already provides image–text pairs augmented with scene-graph structure.
> - All structured supervision in RACA-CLIP— region crops, region description and `⟨subject, relation, object⟩` triplets is obtained by automatically processing these existing annotations, plus a one-time offline LLM pass over the captions.
> - No additional human labels are collected and we do not annotate any new images.
>
> In the revised paper, we will change:
>
> > “scene graph annotated datasets are leveraged for structured supervision without additional labeling”
>
> to:
>
> > “We leverage existing annotated scene-graph datasets (GBC) and use their annotations for getting region crops, region description, and triplet-level supervision, **without collecting any extra manual labels or annotating new images**.”
>
> ---
>
> **(2) Novelty compared to [1,2,3]**
>
> We agree that the high-level idea of using structured annotations to improve multimodal models is not new. Our contribution is **how** we bring this structure into a standard CLIP-style dual encoder in a **purely positive, loss-level way**, while keeping the architecture and inference interface unchanged.
>
> Below we summarize the main differences.
>
>
> ---
> ### (2.1) Type and source of structural supervision
>
>
> - **GLIP [3]** is a detection/grounding model trained on box+label detection datasets with a unified fusion backbone. It does **not** use scene graphs and does **not** preserve a CLIP-like dual-encoder interface.
>
> - **Structure-CLIP [1]** constructs **text-side** scene graphs by parsing captions and uses them to:
>   - build hard negative captions, and
>   - add a Knowledge-Enhanced Encoder branch on the text side.
>   Structure is injected only through the caption graph and an extra text encoder, it will again suffer from the problem of information imbalance between modalities; **visual regions are not directly supervised as separate aligned entities.**
>
> - **Herzig et al. [2] (SGVL)** use Visual Genome image–scene-graph pairs to train **scene-graph prediction heads** on the vision side: object and relation “tokens” are matched to graph nodes/edges with a DETR-style loss, and graph-based captions are used to generate positive and negative examples.
>
> By contrast, **RACA-CLIP**:
>
> - Uses GBC image–scene-graph pairs to construct:
>   - **region–span pairs** (cropped boxes ↔ local text spans), and
>   - `⟨subject, relation, object⟩` **triplets**,
> - And supervises **both** directly in the **joint image–text embedding space** via contrastive losses, using only the existing CLIP encoders.
>
> The goal is to **densify the textual signal relative to the image** (multiple localized captions per image) and address the known imbalance between rich images and sparse global captions, rather than adding graph prediction heads or only restructuring the caption.
>
> ---
> ### (2.2) How compositionality is enforced: ( ours, no hard negatives, no new modules)
>
> Many prior works enforce compositionality through **semantic hard negatives**:
>
> - Structure-CLIP [1] and NegCLIP rely heavily on hard negative captions (scene-graph-guided swaps or word-level corruptions).
> - SGVL [2] similarly uses graph-based negative captions in addition to token-level supervision.
>
> These perturbation-based schemes are powerful, but they are:
>
> - tightly coupled to specific corruption patterns (often similar to SugarCrepe), and
> - can over-specialize to benchmarks that mirror those patterns.
>
> In contrast, **RACA-CLIP**:
>
> - Uses **no semantic hard negatives at all**.
> - Scene graphs are used only to define **positive structural constraints**:
>   - an **IoU-weighted multi-positive region–span loss** (multiple overlapping boxes and their spans are positives; non-overlapping intra-image regions are masked), and
>   - a **relation-aware image–triplet loss** over `⟨subject, relation, object⟩`.
> - Negatives remain the **standard in-batch CLIP negatives**, exactly as in vanilla CLIP.
>
> To the best of our knowledge, this is the **first** work to improve compositionality in a CLIP-style dual encoder using structured supervision **without explicit hard-negative construction and without adding new architectural components.**
>
> Continuing in the next official comment ..

---

> ### Author Response · Authors · 2025-11-24
> **Response to reviewer Upu5 : Related work, Evaluation and Novelty**
>
> ----
> ### (2.3) Architecture and deployability
>
> - **GLIP [3]** replaces the dual encoder with a detection/grounding backbone.
> - **SGVL [2]** introduces additional scene-graph tokens with separate `{Q, K, V, MLP}` parameters in every layer.
> - **Structure-CLIP [1]** adds a Knowledge-Enhanced text encoder branch, introducing additional params.
>
> By contrast, **RACA-CLIP**:
>
> - Keeps the CLIP dual encoder **completely unchanged**: no new tokens, heads, encoders, or architectural branches.
> - Encodes all structural information **in the losses** on top of the standard `encode_image` / `encode_text` outputs.
> - At inference, requires only **raw images and text** (no scene-graph parser, no SG prediction), so RACA-CLIP is a **true drop-in replacement** for CLIP in existing pipelines.
>
> ---
> ### (2.4) Representation-level analysis, not just task metrics
>
> Most prior structured works [1,2,3,4] primarily report task accuracies (e.g., SugarCrepe, Winoground, etc), but provide limited/No analysis of **how the embedding geometry itself changes**.
>
> In addition to benchmark scores, we explicitly analyze the learned representation on **UT-Zappos-50K** and show that RACA-CLIP:
>
> - increases classification accuracy, and
> - reduces cross-class similarity and hubness for both objects and attributes,
>
> indicating **cleaner separation and better structured alignment** in the embedding space. This supports the view that our purely positive, structure-aware objectives genuinely reshape the representation, rather than simply learning to detect a particular family of synthetic negatives.
>
> **Summary of novelty.** Our contribution lies in:
>
> - A **structure-aware, purely positive loss design** for a standard CLIP dual encoder,
> - Using GBC region and triplet supervision to **densify alignment without hard negatives**,
> - **No architectural changes** and **no scene-graph requirements at inference**, and
> - **Representation-level evidence** (UT-Zappos, MMVP-VLM, SugarCrepe++) that the encoder becomes more compositionally structured.
>
> We will state these distinctions more clearly in the method and related-work sections.
>
> ---
>
> **(3) “Incomplete evaluation”: Winoground, MMVP, SugarCrepe++**
>
> We agree that broader evaluation is important. In response, we have run additional experiments on **MMVP-VLM**, **Winoground**, and **SugarCrepe++**.
>
> ### (3.1) MMVP-VLM (fine-grained VQA-style benchmark)
>
> We evaluate RACA-CLIP on MMVP-VLM, which frames fine-grained multimodal understanding as a multiple-choice VQA task. The accuracies of different CLIP variants are:
>
> **Table – MMVP-VLM: Compositional accuracy (%) across visual patterns.**
>
> | Method             | Backbone | Res | Orientation & Direction | Presence of Specific Features | State & Condition | Quantity & Count | Positional & Relational Context | Color & Appearance | Structural Characteristics | Texts | Viewpoint & Perspective | Avg.  |
> |--------------------|:--------:|:---:|:-----------------------:|:-----------------------------:|:-----------------:|:----------------:|:--------------------------------:|:-------------------:|:--------------------------:|:-----:|:------------------------:|:-----:|
> | CLIP               | ViT-B/32 | 224 | 26.7                    | 6.7                           | 40.0              | 6.7              | **26.7**                         | 66.7                | 33.3                       | **26.7** | 13.3                     | 27.4  |
> | NegCLIP            | ViT-B/32 | 224 | 20.0                    | 13.3                          | 46.7              | 13.3             | 20.0                             | 46.7                | 20.0                       | 26.7 | 26.7                     | 25.9  |
> | TripletCLIP        | ViT-B/32 | 224 | 6.7                     | 13.3                          | 40.0              | 6.7              | 13.3                             | 53.3                | 13.3                       | 13.3 | 20.0                     | 20.0  |
> | CE-CLIP            | ViT-B/32 | 224 | 6.7                     | 20.0                          | 46.7              | **13.3**         | 20.0                             | 53.0                | 20.0                       | 20.0 | 13.3                     | 23.7  |
> | *RACA-CLIP (Ours)* | ViT-B/32 | 224 | **26.7**                | **20.0**                      | **60.0**          | 6.7              | 20.0                             | **80.0**            | **40.0**                   | 13.3 | **33.3**                 | **33.3** |
>
>
> RACA-CLIP thus yields a **+5.93 absolute (≈21% relative)** improvement over CLIP, and outperforms NegCLIP, TripletCLIP, and CE-CLIP, indicating improved fine-grained reasoning in a realistic, question-answering setting.
>
>
> Continuing in the next official comment ..

---

> ### Author Response · Authors · 2025-11-24
> **Response to reviewer Upu5 : Related work, Evaluation and Novelty**
>
> ### (3.2) Winoground
>
> We also evaluate on Winoground, which is designed to test difficult compositional disambiguation. Under the same CLIP-like backbone and protocol, we obtain:
>
> **Table – Winoground: Accuracy (%) on text, image, group, and Avg.**
>
> | Method             | Text        | Image        | Group        | Avg.         |
> |--------------------|:-----------:|:------------:|:------------:|:------------:|
> | CLIP               | 30.7 | 10.5         | 8.0   | 16.4  |
> | NegCLIP            | 30.5        | 10.5         | 8.0   | 16.3         |
> | TripletCLIP        | 26.8        | **11.2**     | 7.0          | 15.0         |
> | CE-CLIP            | 18.5        | **11.2**     | 5.0          | 11.6         |
> | *RACA-CLIP (Ours)* | **34.8**    | 11.0  | **9.2**      | **18.3**     |
>
>
> RACA-CLIP attains the **best Winoground score** among these CLIP variants, again **without using hard negatives**.
>
> We will add both MMVP-VLM and Winoground results to the revised manuscript (with concise discussion in the main text and full tables in the appendix).
>
> ### (3.3) Additional benchmark: SugarCrepe++
>
> Beyond SugarCrepe and MMVP-VLM, we evaluate on **SugarCrepe++**, which focuses on **multi-positive, paraphrastic compositional understanding**:
>
> **Table – SugarCrepe++: Compositional accuracy (%) on replace and swap perturbations.**
>
> | Method             | Replace Obj | Replace Attr | Replace Rel | Swap Obj | Swap Attr | Avg.  |
> |--------------------|:-----------:|:------------:|:-----------:|:--------:|:---------:|:-----:|
> | CLIP               | 87.89       | 68.02        | 50.14       | 37.95    | 48.19     | 58.4  |
> | NegCLIP            | 89.58       | 69.54        | 52.43       | **54.69** | **58.40** | 64.9  |
> | TripletCLIP        | 86.98       | 71.82        | **62.23**   | 39.59    | 48.49     | 61.8  |
> | CE-CLIP            | 81.84       | 62.43        | 53.12       | 40.00    | 40.39     | 55.6  |
> | *RACA-CLIP (Ours)* | **91.88**   | **74.49**    | 59.53       | 44.48    | 55.10     | **65.1** |
>
>
> - Each image is paired with **two lexically different but semantically correct captions** and one hard negative caption that is lexically similar but semantically incorrect.
> - The benchmark evaluates: **image→text retrieval**, where both positives should rank above the hard negative, and
>
> On SugarCrepe++:
>
> - RACA-CLIP achieves the **best performance on image→text retrieval**, outperforming NegCLIP, CE-CLIP, and TripletCLIP. This suggests that our region- and triplet-aware positive supervision naturally supports **multiple lexically diverse but semantically consistent captions per image**, matching real-world usage where different users describe the same scene differently.
>
> These results complement SugarCrepe and MMVP-VLM and further support our claim that RACA-CLIP improves cross-modal compositional representations **without relying on hard negatives or architectural changes**. We will add SugarCrepe++ results to the revised manuscript.
>
> ---
>
> **(4) Related work coverage and SugarCrepe baselines**
>
> We agree that our related work discussion and baselines around SugarCrepe can be made clearer and more complete.
>
> ### (4.1) Expanded discussion of [1,2,3]
>
> In the revision, we will explicitly discuss:
>
> - **Structure-CLIP [1]:**
>   Parses captions into text-side scene graphs, builds semantic hard negatives, and adds a Knowledge-Enhanced text encoder branch. Structure is injected via the caption graph and extra encoder, with no direct region-level alignment in the shared embedding.
>
> - **SGVL / Herzig et al. [2]:**
>   Uses Visual Genome to define scene-graph tokens on the vision side, with DETR-style matching for object/relation nodes and graph-based positive/negative captions.
>
> - **GLIP [3]:**
>   A detection/grounding model trained on detection datasets with a unified fusion backbone; it does not operate as a CLIP-style dual encoder and does not use scene graphs in the same way.
>
> We will contrast these with **RACA-CLIP**, which:
>
> - does **not** introduce new encoders, scene-graph heads, or knowledge-enhanced branches,
> - does **not** construct semantic hard negatives, and
> - uses GBC to define **positive IoU-weighted region–span and image–triplet losses** directly in the shared CLIP embedding space, keeping the dual-encoder architecture intact.
>
> ### (4.2) SugarCrepe baselines and practical constraints
>
> We agree that methods such as CE-CLIP [4] and Structure-CLIP [1] are relevant for SugarCrepe-style evaluation, but there are practical issues:
>
> - Structure-CLIP [1] does **not** report SugarCrepe results, so there are no official numbers to reuse.
> - It also does **not** release checkpoints, only code; re-training their full system just to obtain SugarCrepe numbers would be expensive and potentially yield mismatched comparisons.
> - SGVL [2] mainly releases BLIP/BLIP2-style models, which are not directly comparable to our ViT-B/32 CLIP fine-tuning under a fixed-backbone protocol.
>
> Continuing in the next official comment ..

---

> ### Author Response · Authors · 2025-11-24
> **Response to reviewer Upu5 : Related work, Evaluation and Novelty**
>
> ### (4.3) Clarifying our position in the literature
>
> We will revise the related work to position **RACA-CLIP** as:
>
> - a **structure-aware, purely positive contrastive fine-tuning** of a standard CLIP dual encoder,
> - using GBC region and triplet supervision to improve compositionality **without hard negatives** and **without architectural changes**, and
> - requiring **no scene-graph machinery at inference**.

---

> > ### Comment · Reviewer_Upu5 · 2025-11-27
> >
> > I appreciate the authors’ detailed responses to my earlier concerns. The new benchmark results help clarify the effectiveness of the method. That said, I still find the approach somewhat limited. The main contribution relies on leveraging existing, richly annotated scene-graph datasets to fine-tune CLIP for improved performance on current benchmarks. While the method uses standard CLIP without additional components and avoids hard-negative mining, these choices do not represent a fundamental departure from prior work—for example, the method can similarly treat positives as negatives for unrelated samples within a batch. It's more like a choice of design rather than difference in the methodology.
> >
> > More importantly, the approach depends heavily on high-quality scene-graph data. Even without extra annotation effort, obtaining such datasets is costly and restricts the scalability of the method.
> >
> > Taking these points into account, I have updated my score to 4.

---

> > > ### Author Response · Authors · 2025-11-30
> > > **Response to reviewer Upu5 :Response on Novelty, Geometric Contributions, and Structured Supervision**
> > >
> > > We thank the reviewer Upu5 for the clarification and for updating their assessment. We would like to address the remaining concerns regarding **methodological novelty** and **dependence on annotated scene-graph data**.
> > >
> > > **(1) On novelty beyond “design choices”**
> > > Our contribution is not simply that we leverage existing scene-graph datasets; it is how this **structured information reshapes the joint vision–language embedding space**. Standard contrastive CLIP training—whether using **in-batch negatives**, caption-only hard negatives, or heuristically swapped captions—optimizes only **one half of the joint metric space**. These approaches are known to separate captions effectively but leave the **image-side compositional geometry underconstrained**, producing the asymmetric behavior highlighted in recent bidirectional VLC studies (e.g., **Miranda et al., NeurIPS 2024**), where **image→text** improves while **text→image** and **fine-grained visual distinctions** remain entangled.
> > >
> > > **RACA-CLIP** introduces **region-aligned positives** and **relation-aware triplet supervision**, which impose **bidirectional, part-level geometric constraints** that classical contrastive formulations cannot express. This is not achievable by simply “treating unrelated positives as negatives”: in-batch negatives do not specify **which region, attribute, or relation** is responsible for the mismatch, so the model has no way to adjust the internal geometry in a **role-sensitive** manner. In contrast, our structured losses explicitly ensure that **object identity**, **attributes**, and **relational roles** occupy **distinct, separable subspaces**, leading to a **qualitatively different embedding geometry**.
> > >
> > > Our **UT-Zappos50k representation diagnostic** further confirms this distinction: while baseline CLIP and hard-negative variants exhibit **conflation between object types and attributes**, **RACA-CLIP** reorganizes the latent space into **cleaner, more orthogonal manifolds**. This geometric restructuring is precisely why RACA-CLIP **generalizes consistently** across **SugarCrepe**, **WhatsUp**, **SugarCrepe++**, and **Zappos** categories—even though the **model architecture remains unchanged**. Such geometric improvements **cannot be reproduced** by alternative batching or negative-sampling strategies.
> > >
> > > **(2) On reliance on scene-graph annotations**
> > > We agree that richly annotated **scene-graph datasets** are nontrivial to collect. Importantly, **RACA-CLIP** uses these annotations **only once during training** as a **structural regularizer**, not as a dependency at inference or for scaling. The value of scene-graph supervision here is not in the quantity of data but in its **granularity**: **object spans** and **relational roles** provide precisely the **unit-level signals** needed to resolve the geometric pathologies noted above. We also show that even **modest coverage** of structured data is sufficient to improve **compositional alignment** across multiple benchmarks without increasing model size or requiring additional **inference-time components**.
> > >
> > > Finally, in contrast to methods relying on **mined hard negatives** or **additional decoding modules**, **RACA-CLIP** keeps the **backbone architecture unchanged** and preserves **CLIP’s scalability**. Structured supervision serves to **correct biases in contrastive alignment** rather than to introduce an ongoing **annotation bottleneck**.
> > >
> > > **(3) Summary**
> > > While the architecture remains **CLIP**, the **training signal** is fundamentally different: **RACA-CLIP** implements **role-aware, region-grounded constraints** that **restructure the geometry** of the embedding space in ways that existing **in-batch** or **hard-negative** strategies cannot. We hope this clarification helps situate the contribution as a **methodological shift** in how **compositional alignment** is enforced rather than a **dataset choice** alone.
> > >
> > > ---
> > >
> > > **Reference:**
> > > **Miranda et al.**, *“BIVLC: Extending Vision–Language Compositionality Evaluation with Text-to-Image Retrieval,”* **NeurIPS 2024** (Datasets and Benchmarks).

---

### Official Review · Reviewer_vCJz · 2025-11-01

**Soundness:** 3
**Presentation:** 3
**Contribution:** 4
**Rating:** 6
**Confidence:** 2

**Summary:**

This paper proposes RACA-CLIP, a structured contrastive framework that enhances CLIP’s compositional reasoning by integrating scene-graph-based supervision. This paper introduces region-level IoU-weighted alignment and relation-aware triplet losses to better capture object–attribute bindings and inter-object relations. Trained on the graph-based captioning dataset, RACA-CLIP achieves large improvements on compositional benchmarks

**Strengths:**

The model demonstrates robust performance on several compositional benchmarks, outperforming CLIP, NegCLIP, and TripletCLIP while maintaining strong zero-shot and retrieval accuracy.

This paper clearly explains the global-only limitation of CLIP and motivates structured alignment with strong intuition and references.

**Weaknesses:**

The method heavily depends on scene-graph annotations and LLM-based triplet extraction, which may introduce noise or inconsistencies and limit scalability to unstructured web data.

The computational cost of extracting scene graphs and aligning multiple fine-grained objectives isn’t discussed. Could you please discuss it?

If the impact on broader downstream tasks, such as visual question answering or image generation, this paper will be more comprehensive.

**Questions:**

Refer to Weaknesses

---

> ### Author Response · Authors · 2025-11-24
> **Response to reviewer vCJz : Scene-Graph Dependence, Compute Overhead, and Downstream Impact**
>
> We thank Reviewer vCJz for the constructive feedback and the opportunity to address these concerns.
>
> ---
> **(1) Dependence on scene-graph and LLM supervision**
>
> We agree that naively requiring scene-graph annotations for arbitrary web images would raise scalability concerns. Our setting is more modest: **we do not create any new scene-graph annotations**, but train on the publicly available **GBC** dataset, which already provides image–text pairs with scene-graph structure. This reuses existing supervision, analogous to CLIP variants that rely on datasets with bounding boxes or region labels, **without adding human annotation cost**.
>
> LLM-based triplets are extracted **once, offline**, by processing captions with Qwen2.5-7B-Instruct + vLLM; generating triplets for the full GBC subset took **≈ 3 hours on 6×V100**, which is negligible compared to CLIP pre-training. At test time, **RACA-CLIP consumes only raw images and text**, exactly like standard CLIP. All our evaluations use datasets without scene graphs or triplet annotations, showing that the structured supervision learned from GBC transfers to unstructured web-style data **without any scene-graph or LLM processing at deployment**. The **structured annotations are used only during training**.
>
> To address the concern about noise, we run a robustness study where we corrupt a fraction of region spans and triplet relations in the scene graphs during training (Table 1). With clean graphs, RACA-CLIP reaches an average accuracy of 46.2 vs 39.3 for CLIP, **averaged across all compositional benchmarks (WhatsUp, MMVP, etc)**. With 10% and 20% corruption. **Most gains are therefore preserved under synthetic noise**, showing that RACA-CLIP remains effective even when the structured supervision is noisy.
>
> **Table 1 – Robustness to scene-graph noise (average).**
>
> |                  | CLIP | RACA-CLIP (0% noise) | RACA-CLIP (10% noise) | RACA-CLIP (20% noise) |
> |------------------|:----:|:--------------------:|:---------------------:|:---------------------:|
> | Avg. Acc. (%)    | 39.3 | **46.2**             | 44.8                  | 43.6                  |
>
> ---
>
>
> **(2) Computational cost of structured supervision and fine-grained objectives**
>
> We **do not run any additional scene-graph extraction**; we simply use the graphs already provided in GBC. The only extra preprocessing is LLM-based triplet extraction, done once offline (≈3 hours on 6×V100). During training, RACA-CLIP uses the **same image and text encoders as CLIP, with no new backbone**. All objectives share a single forward pass: we batch global images and region crops into one `encode_image` call, and global captions, region descriptions, and triplet texts into one `encode_text` call, then compute global/region/triplet losses on shared embeddings. Thus, the **backbone parameter count and per-forward FLOPs at inference are unchanged**; the additional cost is only from multiple views and extra loss terms during training. On GBC1M with ViT-B/32 and 6×V100, this yields a **moderate increase in training time** compared to a CLIP-only baseline on the same dataset, but remains within a practical fine-tuning budget and is far from re-training a foundation model. **Crucially, there is no extra inference cost** at test time.
>
> ---
> **(3) Impact on broader downstream tasks (VQA and image generation)**
>
> Our main goal is to improve compositional representations in CLIP-style encoders and study their effect on compositional benchmarks. In response to the reviewer, we additionally evaluate RACA-CLIP on **MMVP-VLM**, which frames fine-grained multimodal understanding as a VQA-style multiple-choice task.
>
> **Table 2 – MMVP-VLM average accuracy.** Compositional accuracy (%) averaged across all visual attribute categories.
> |                      | CLIP | NegCLIP | TripletCLIP | CE-CLIP | _RACA-CLIP (Ours)_ |
> |----------------------|:----:|:-------:|:-----------:|:-------:|:------------------:|
> | MMVP-VLM Acc. (%)    | 27.40 | 25.92  | 20.00       | 23.70   | **33.33**          |
>
> Thus, **RACA-CLIP improves over vanilla CLIP by +5.93 absolute points (≈21% relative) and outperforms all other structured variants**. Unlike several prior structured/compositional approaches that rely on hard-negative mining, **RACA-CLIP does not use explicit hard negatives**, yet achieves the best performance, suggesting that region- and triplet-aware positive supervision is sufficient to learn strong compositional structure without negative-mining schemes.
> Architecturally, **RACA-CLIP is just a fine-tuned CLIP encoder with the same `encode_image` / `encode_text` interface**, so it can be used as a **drop-in backbone** in existing pipelines that rely on CLIP features. For generative models, it can serve as an external alignment/scoring model, while **replacing the internal text encoder of diffusion models and jointly fine-tuning the generator is left for future work**.
>
> ---
> The detailed table (MMVP-VLM, etc) will be in the appendix of the revised PDF.

---

### Author Response · Authors · 2025-12-01
**General response to reviewers and AC (Summary of Revisions in Response to Reviewers Feedback)**

We thank all four reviewers for their detailed and constructive feedback. We have uploaded a revised PDF in which all substantive additions and clarifications are highlighted in **blue** for ease of inspection. Below, we summarize how the main concerns were addressed and where the corresponding changes appear.

**1. Novelty, hard negatives, and relation to structure-aware prior work**
*(Asked by Reviewers: Upu5, NBkA, vCjz, ftKs)*

- Expanded the comparison to Structure-CLIP, SGVL, and related methods in **Appendix B.4**, clarifying:
  - where structure is injected (text-only vs. dual encoder),
  - how prior approaches rely on hard negatives (whereas RACA-CLIP does not), and
  - architectural differences (previous methods introduce new modules, while RACA-CLIP keeps the CLIP dual encoder unchanged).

- Added a new subsection **“Why RACA-CLIP Avoids Hard-Negative–Driven Training” in Appendix E.10**, explaining:
  - the information imbalance in CLIP-style training,
  - the asymmetric geometry induced by hard-negative–driven objectives, and
  - how our region- and triplet-based positive supervision yields a more symmetric and visually grounded embedding space.


**2\. Evaluation breadth beyond SugarCrepe**\
*(Reviewers : Upu5, vCjz, ftKs)*

-   Extended the **fine-grained understanding analysis** with **MMVP-VLM** results in the **main paper, Section 4.7**.

-   Added **Winoground** results in **Appendix E.5** and **SugarCrepe++** results in **Appendix E.6**, providing a broader view of compositional and paraphrastic robustness.

**3\. Sensitivity to scene-graph noise and annotation quality**\
*(Reviewers : vCjz, NBkA, ftKs)*

-   Added a **robustness-to-noise study in Section 4.8**, where 10--20% of region spans and triplet relations are corrupted during training.

-   The results show smooth degradation while remaining above CLIP, indicating that scene graphs act as a **soft inductive bias** rather than a brittle requirement.

**4\. Causal evidence: data vs. structured losses**\
*(Reviewers : NBkA, ftKs)*

-   Introduced a **CLIP-GBC (global)** baseline, as well as **region-only** and **triplet-only** variants.

-   Reported in **Appendix E.8**, these ablations show that both structured losses independently improve over CLIP-GBC and their combination yields the full RACA-CLIP gains, providing causal evidence that the structured objectives themselves drive the improvements.

**5\. Computational overhead and practicality**\
*(Reviewers : vCjz, ftKs)*

-   Added **"Structured Supervision and Computational Overhead" in Appendix D.4**, clarifying that:

    -   the CLIP backbone architecture and per-forward FLOPs remain unchanged,

    -   extra training cost comes from multi-view batching and structured losses,

    -   LLM-based triplet extraction is a **one-time offline** step, and

    -   **inference cost (parameters, FLOPs, latency) matches vanilla CLIP**.

**6\. Scaling with backbone capacity**\
*(Reviewer : NBkA)*

-   Added **ViT-L/14** experiments in **Appendix E.9**, showing improvements from 39.3→46.2 (ViT-B/32) and 43.1→48.3 (ViT-L/14), confirming that benefits persist and even strengthen as backbone capacity increases.

**7\. Qualitative evidence**\
*(Reviewers : NBkA, ftKs)*

-   Added additional dense-caption alignment examples in **Appendix E.11**, comparing RACA-CLIP to NegCLIP and TripletCLIP on complex multi-object scenes, visually illustrating the improved compositional grounding suggested by our quantitative results.

All of these changes are highlighted in **blue** in the revised manuscript and are explicitly tied to the concerns raised in the reviews. We hope it is clear that we have carefully incorporated the feedback and that the revised manuscript more convincingly addresses concerns about novelty, robustness, evaluation breadth, and practical applicability.

---
**Impact Summary**
------------------

RACA-CLIP is, to our knowledge, **the first CLIP-style dual-encoder approach that achieves strong compositional and geometric alignment **without relying on any form of hard-negatives**. Instead of reshaping the *text* manifold through synthetic caption perturbations, RACA-CLIP strengthens the embedding space via region-level and relation-level positive constraints---**leading to a more balanced and structurally coherent multimodal geometry** that better reflects object-, attribute-, and relation-level distinctions.

By avoiding hard negatives, the method becomes less brittle and does not depend on templated or LLM-generated negatives, while remaining practical because it uses only GBC’s region annotations and a one-time triplet extraction. UT-Zappos-50K, noise-robustness, and causal studies further show that RACA-CLIP’s gains stem from **genuine structural alignment** rather than data artifacts.

We hope these revisions and additional analyses provide a clearer picture of the contribution and its potential impact on future multimodal representation learning.

---

### Meta-Review · Area_Chair_xKKr · 2026-01-07

**Summary:**

The paper proposes using a semantic graph to improve CLIP. Yet, there is a question about the novelty as prior works also used semantic graphs and I did not observe an adequate response for that. Moreover, there is a question about the scalability of using a semantic graph. The use of semantic graph proved a very useful supervision that is not present in other works. Understanding that adding this information is an important contribution. Yet, given that this part is not novel, I don't think the paper can be accepted given the other problems raised by the reviewers.

**Reviewer Concerns:**

The reviewers raised questions about novelty, comparison and relation to prior work and asked for more evaluations. The authors explained some of the unclear things and added some new comparisons but the novelty w.r.t prior work was not clarified in a satisfactory manner.

**Reviewer Scores:**

Some reviewers would increase their score slightly but not in a satisfactory manner that will merit acceptance.

---

### Decision · Program_Chairs · 2026-01-26

Reject